# The Mars1 kinase confers photoprotection through signaling in the chloroplast unfolded protein response

Karina Perlaza[1,2], Hannah Toutkoushian[1,2], Morgane Boone[1,2], Mable Lam[1,2], Masakazu Iwai[3], Martin C Jonikas[4], Peter Walter[1,2]*, Silvia Ramundo[1,2]*

[1]Howard Hughes Medical Institute, University of California, San Francisco, San Francisco, United States; [2]Department of Biochemistry and Biophysics, University of California, San Francisco, San Francisco, United States; [3]Molecular Biophysics and Integrated Bioimaging Division, Lawrence Berkeley National Laboratory, Berkeley, United States; [4]Department of Molecular Biology, Princeton University, Princeton, United States

**Abstract** In response to proteotoxic stress, chloroplasts communicate with the nuclear gene expression system through a chloroplast unfolded protein response (cpUPR). We isolated *Chlamydomonas reinhardtii* mutants that disrupt cpUPR signaling and identified a gene encoding a previously uncharacterized cytoplasmic protein kinase, termed Mars1—for mutant affected in chloroplast-to-nucleus retrograde signaling—as the first known component in cpUPR signal transmission. Lack of cpUPR induction in *MARS1* mutant cells impaired their ability to cope with chloroplast stress, including exposure to excessive light. Conversely, transgenic activation of cpUPR signaling conferred an advantage to cells undergoing photooxidative stress. Our results indicate that the cpUPR mitigates chloroplast photodamage and that manipulation of this pathway is a potential avenue for engineering photosynthetic organisms with increased tolerance to chloroplast stress.

*For correspondence:
peter@walterlab.ucsf.edu (PW);
silvia@walterlab.ucsf.edu (SR)

Competing interests: The authors declare that no competing interests exist.

## Introduction

In photosynthetic eukaryotes chloroplasts fulfill many essential functions such as photosynthetic conversion of light into chemical energy, synthesis of essential amino acids, fatty acids and other secondary metabolites. Moreover, they act as signaling platforms during plant development and stress adaptation, as they can alter the expression of thousands of nuclear genes and influence many cellular activities that are key to plant performance (*Chan et al., 2016*). Selective impairment of protein homeostasis in chloroplasts triggers the chloroplast unfolded protein response (cpUPR), a conserved organelle quality control pathway (*Ramundo et al., 2014*; *Llamas et al., 2017*). Akin to unfolded protein responses operating from the endoplasmic reticulum (ER) and mitochondria (*Walter and Ron, 2011*; *Shpilka and Haynes, 2018*), the cpUPR invokes comprehensive transcriptional changes thought to mitigate an increased burden of proteotoxicity in the organelle. As such, the cpUPR comprises the selective up-regulation of nuclear encoded chloroplast-localized small heat shock proteins, chaperones, proteases, and proteins involved in chloroplast membrane biogenesis. Furthermore, other pathways, such as autophagy and sulfur uptake are activated to mitigate general cellular stress caused by chloroplast metabolic dysfunctions (*Ramundo et al., 2014*).

In the single-celled alga *Chlamydomonas reinhardtii*, the cpUPR is induced after either inactivation of the Clp protease, which degrades misfolded chloroplast proteins in the organelle's stroma (*Figure 1A*), or exposure to higher than normal light intensity (high light 'HL'), which causes protein damage through the production of reactive oxygen species in the chloroplast

**eLife digest** Life on Earth crucially depends on photosynthesis, the process by which energy stored in sunlight is harnessed to convert carbon dioxide into sugars and oxygen. In plants and algae, photosynthesis occurs in specialized cellular compartments called chloroplasts. Inside chloroplasts, complex molecular machines absorb light and channel its energy into the appropriate chemical reactions. These machines are composed of proteins that need to be assembled and maintained. However, proteins can become damaged, and when this occurs, they must be recognized, removed, and replaced.

When exposed to bright light, the photosynthetic machinery is pushed into overdrive and protein damage is accelerated. In response, the chloroplast sends an alarm signal to activate a protective system called the "chloroplast unfolded protein response", or cpUPR for short. The cpUPR leads to the production of specialized proteins that help protect and repair the chloroplast.

It was not known how plants and algae evaluate the level of damaged proteins in the chloroplast, or which signals trigger the cpUPR. To address these questions, Perlaza et al. designed a method to identify the molecular components of the alarm signal. These experiments used specially engineered cells from the algae *Chlamydomonas reinhardtii* that fluoresced when the cpUPR was activated. Perlaza et al. mutagenized these cells – that is, damaged the cells' DNA to cause random changes in the genetic code. If a mutagenized cell no longer fluoresced in response to protein damage, it indicated that communication between protein damage and the cpUPR had been broken. In other words, the mutation had damaged a piece of DNA that encoded a protein critical for activating the cpUPR.

These experiments identified one protein – which Perlaza et al. named Mars1 – as a crucial molecular player that is required to trigger the cpUPR. Algal cells with defective Mars1 were more vulnerable to chloroplast damage, including that caused by excessive light.

These discoveries in algae will serve as a foundation for understanding the mechanism and significance of the cpUPR in land plants. Perlaza et al. also found that mild artificial activation of the cpUPR could preemptively guard cells against damaged chloroplast proteins. This suggests that the cpUPR could be harnessed in agriculture, for example, to help crop plants endure harsher climates.

(*Ramundo et al., 2014*). Similarly, in higher plants, mutants with constitutively reduced levels of the Clp and FtsH proteases selectively upregulate the expression of chloroplast chaperones, such as Cpn60, Hsp70, Hsp90, Hsp100 (*Llamas et al., 2017*; *Zybailov et al., 2009*; *Sjögren et al., 2004*; *Rudella et al., 2006*; *Dogra et al., 2019*). However, the mechanism by which chloroplast proteotoxic stress is monitored and how the signal is transmitted from the organelle to the nucleus has remained unknown.

## Results and discussion

To identify molecular components that mediate cpUPR signaling, we carried out a forward genetic screen in *C. reinhardtii*. To this end, we developed a high-throughput plate-based imaging assay to detect transcriptional activation of cpUPR target genes. In brief, we engineered a reporter strain, in which a truncated promoter and 5' untranslated region of *VIPP2* (*Nordhues et al., 2012*), an early-responsive and highly selective cpUPR target gene (*Ramundo et al., 2014*), was fused to the coding sequence of yellow fluorescent protein (YFP) (*Figure 1A*). The reporter cells also contained a vitamin-toggled chimeric promoter/riboswitch that allowed efficient shut-down of ClpP1 expression upon addition of two vitamins (Vit), thiamine and vitamin $B_{12}$, to the medium (*Ramundo et al., 2014*; *Croft et al., 2007*; *Helliwell et al., 2014*). ClpP1 is an essential chloroplast-encoded subunit of the Clp protease (*Kuroda and Maliga, 2003*; *Huang et al., 1994*). Such design allowed us to trigger the cpUPR by replica-plating onto media containing Vit, yielding a quantitative readout of cpUPR activation (*Figure 1B–C*, *Figure 1—figure supplement 1A–B*). Immunoblotting confirmed that the reporter strain induced YFP with comparable kinetics to those of the Vipp2 protein induction upon ClpP1 repression (*Figure 1C*). As expected, HL, representing a more physiological stress, similarly induced expression of YFP (*Figure 1—figure supplement 1C*).

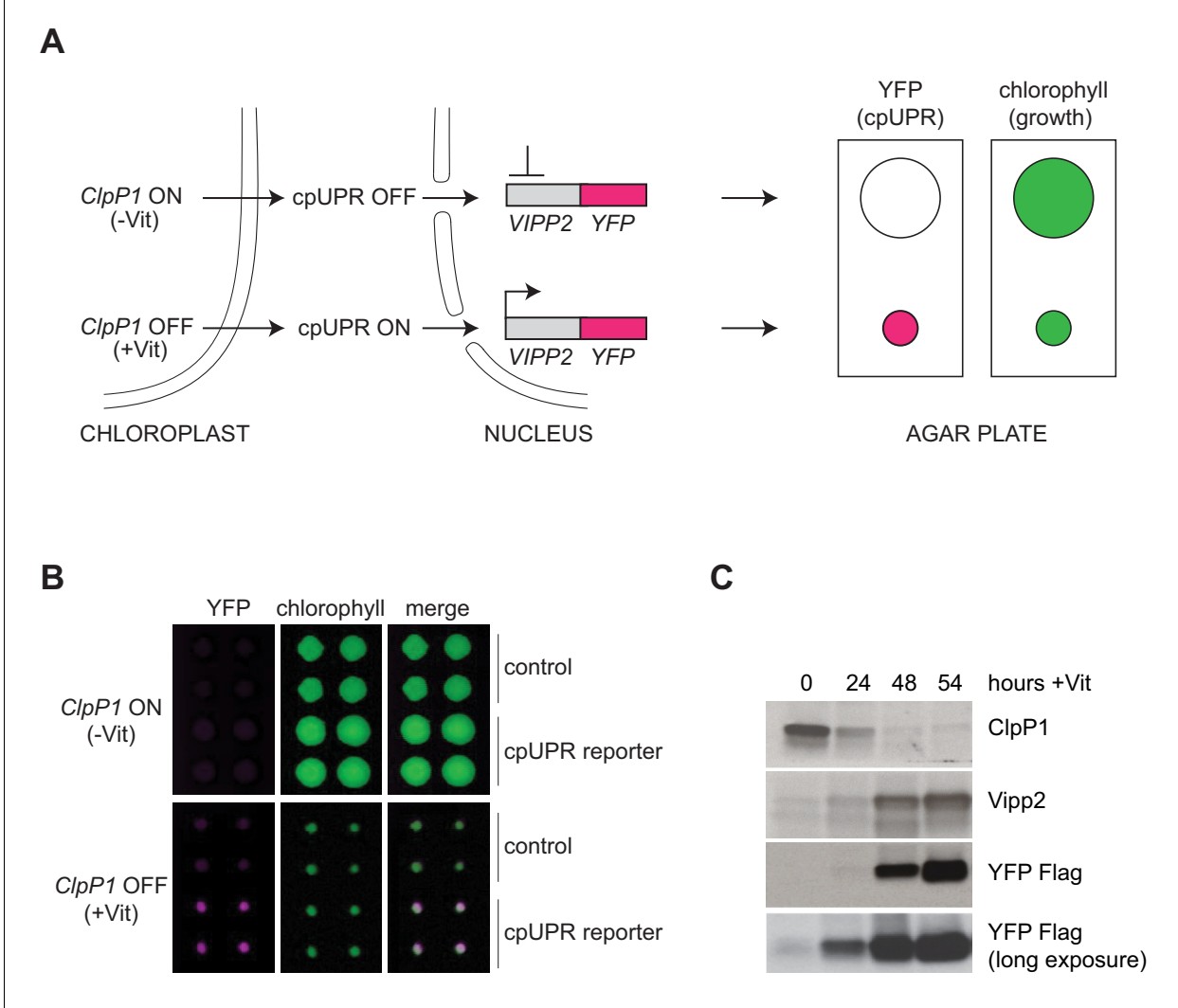

**Figure 1.** Reporter strain for high-throughput screening for cpUPR mutants. (**A**) Schematic of cpUPR regulation in the reporter strain. Under homeostatic conditions (-Vit), chloroplast ClpP1 is expressed and the cpUPR pathway is OFF, as indicated by the lack of *VIPP2* expression; upon ClpP1 depletion (+Vit), the cpUPR is induced leading to *VIPP2* expression. The cells contain an inducible reporter gene consisting of the *VIPP2* promoter fused to the coding sequence of YFP tagged with a 3x-Flag epitope at its C-terminus. When the reporter gene is activated, YFP fluorescence is induced (magenta circles), and the reporter strain's growth is inhibited, as indicated by the smaller colony size (green circles) measured in the chlorophyll-imaging channel. (**B**) Plate-based real-time imaging assay to detect cpUPR activation. Four technical replicates of control cells (containing only the ClpP1-repressible system) and of cpUPR reporter cells (additionally containing the *YFP* reporter gene) were imaged after 6 days of growth on agar plates under ClpP1-permissive or ClpP1-nonpermissive conditions (-/+Vit, respectively). Induction of the YFP fluorescence is observed exclusively in the reporter strain replicates, while growth inhibition is observed in both control and cpUPR reporter strains in ClpP1-nonpermissive conditions. (**C**) Immunoblots of reporter cell extracts upon ClpP1 repression (+Vit) for 0, 24, 48, and 54 hr were probed with anti-ClpP1, anti-Vipp2, and anti-Flag antibodies.

The online version of this article includes the following figure supplement(s) for figure 1:

**Figure supplement 1.** Design of a reporter strain for high-throughput detection of the cpUPR signaling in *C. reinhardtii*.

For mutagenesis, we randomly integrated a cassette expressing paromomycin resistance into the reporter cells. We isolated colonies and re-arrayed them robotically in 384-well agar plates and then replicated them onto plates without Vit (ClpP1-permissive) or with Vit (ClpP1-nonpermissive). We screened 10,709 insertional mutants for YFP intensity and colony size at 2 and 6 days after replica plating (*Figure 2A–B*, *Figure 2—source data 1*). We next scored mutants carrying cpUPR-silencing mutations by their lack of YFP fluorescence in the ClpP1-nonpermissive condition (+Vit) and those

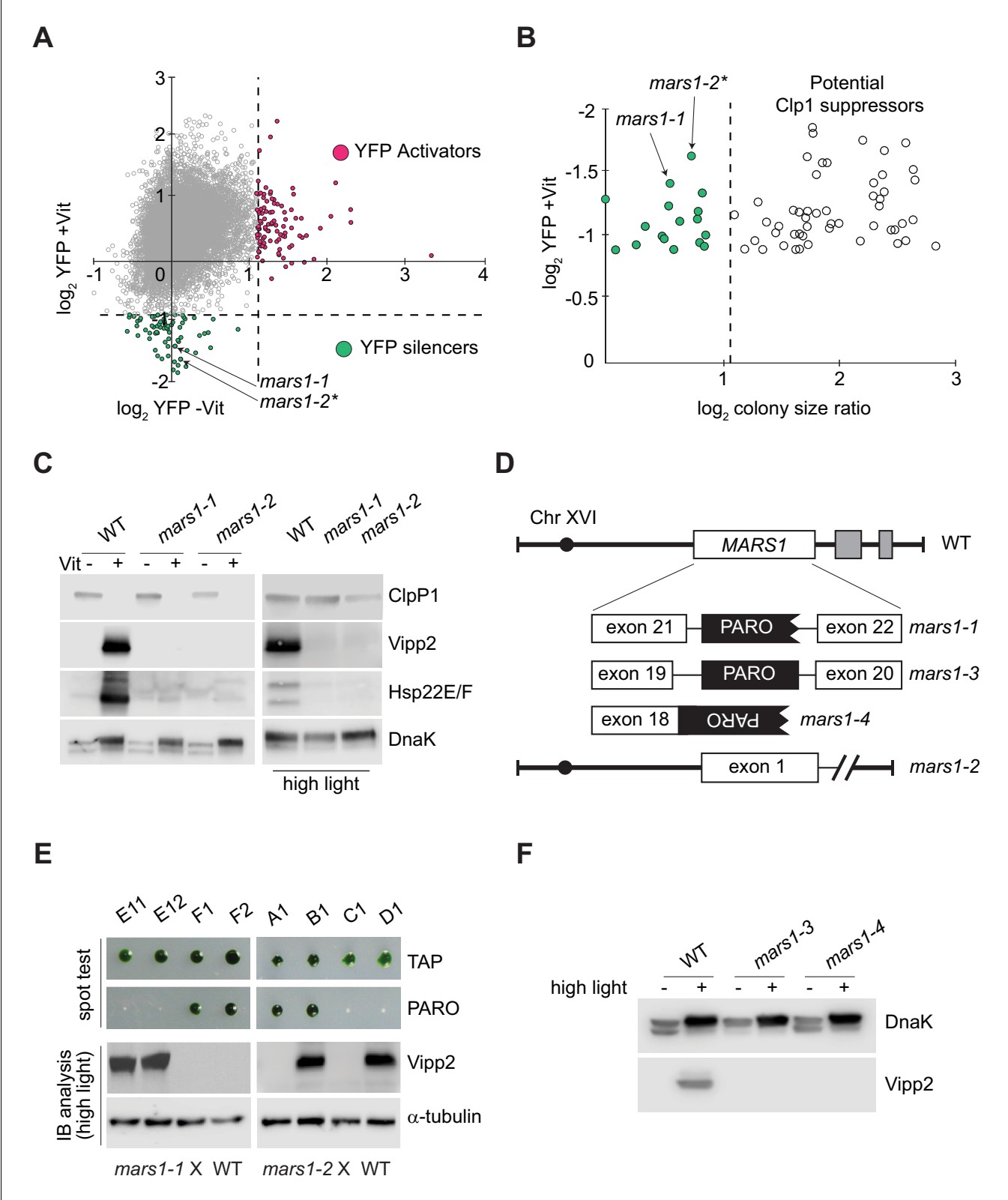

**Figure 2.** Genetic screen identifying *mars1-1* and *mars1-2*. (**A**) Scatter plot of changes in YFP fluorescence for each mutant relative to reporter cells in ClpP1-permissive (–Vit) and ClpP1-nonpermissive (+Vit) conditions. Mutants exhibiting YFP fluorescence at least three standard deviations lower or higher than the mean (dotted lines) were categorized as YFP silencers and activators. Positions of the cpUPR silencers *mars1-1* and *mars1-2\** are indicated (for details on *mars1-2\** refer to the note in Supplementary Materials). (**B**) Scatter plot of colony size ratio over fold-changes in YFP

*Figure 2 continued on next page*

Figure 2 continued

fluorescence for each mutant relative to reporter cells under ClpP1-nonpermissive conditions (+Vit). Colony size ratio was calculated as a fold-change of the colony area between 2 and 6 days after plating. The average colony size increase is indicated by the dashed line. Colonies that increased in size more than average represent potential suppressors of ClpP1 repression. (C) Immunoblot analysis of cpUPR reporter cell (WT) extracts, mars1-1 and mars1-2 cells grown in ClpP1-permissive or ClpP1-nonpermissive conditions (-/+Vit, respectively) or exposed to high light, using antibodies against ClpP1, Vipp2, Hsp22E/F and DnaK (loading/stress control). (D) Diagram of MARS1 indicating the insertion site of the mutagenic cassette (PARO) in each respective MARS1 mutant allele. Gray boxes indicate neighboring genes and the interrupted line a deletion. (E) Analysis of representative meiotic tetrads from backcrosses of mars1-1 and mars1-2 to WT (CC-124) (E11-F2 and A1-D1 correspond to the plate coordinates in *Figure 2—figure supplement 2B*, *Figure 2—figure supplement 3B*). Tetrads were spotted on acetate agar (TAP) and on acetate agar supplemented with paromomycin (PARO). Samples prepared from the strains grown under HL were immunoblotted with antibodies against Vipp2 and α-tubulin (loading control). (F) Immunoblot samples prepared from WT, mars1-3 and mars1-4 cells grown under control or HL conditions were probed with antibodies against Vipp2 and DnaK as a loading/stress control.

The online version of this article includes the following source data and figure supplement(s) for figure 2:

**Source data 1.** Scoring mutants via YFP and area measurements.
**Figure supplement 1.** The genetic screen yields two cpUPR silencing mutants, mars1-1 and mars1-2.
**Figure supplement 2.** Genetic analysis of mars1-1.
**Figure supplement 3.** Genetic analysis of mars1-2.
**Figure supplement 4.** Genetic analysis of mars1-3 and mars1-4.

carrying cpUPR-activating mutations by their constitutive YFP fluorescence even in the ClpP1-permissive condition (-Vit) (*Figure 2A*).

We focused on the cpUPR-silencing mutants that exhibited YFP levels at least three standard deviations lower-than-average YFP fluorescence of all mutants subjected to ClpP1 repression (*Figure 2A*). This non-saturating screen yielded 68 mutants, of which 51 gave rise to colonies larger than those of the parental cpUPR reporter strain on ClpP1-nonpermissive plates (*Figure 2B*), suggesting that they impaired ClpP1 repression (e.g., *Figure 2—figure supplement 1A–C*). Of the remaining 17 mutants, we excluded 15 based on immunoblot analyses that suggested that these mutants contain an insertion affecting only the YFP reporter (e.g., false positive shown in *Figure 2—figure supplement 1D*). The two remaining mutants exhibited a complete defect in the induction of the cpUPR target genes upon ClpP1 repression, demonstrated by their lack of Vipp2 and Hsp22E/F, another strongly induced protein during the cpUPR (*Ramundo et al., 2014*; *Rütgers et al., 2017*; *Figure 2C*). Vipp2 and Hsp22E/F induction was also impaired during HL, further underscoring the two mutants' cpUPR-silencing phenotype (*Figure 2C*). As we show below, the two mutants are allelic, both bearing disruptions in Cre16.g692228 (*Figure 2D*). We henceforth refer to this gene as MARS1 (for mutants affecting retrograde signaling) and the mutants as mars1-1 and mars1-2.

MARS1 is a previously uncharacterized nuclear gene located at the end of chromosome XVI (*Figure 2D*). It encodes a large protein with no known motifs but a predicted serine/threonine kinase domain toward its C-terminus.

In the case of mars1-1, the gene was disrupted by insertion of the mutagenic cassette in intron 21 (*Figure 2D*). Both tetrad and random spore analyses of WT x mars1-1 backcrosses confirmed that the insertion of the cassette in MARS1 (conferring paromomycin resistance) co-segregated with the cpUPR-silencing phenotype (*Figure 2E*, *Figure 2—figure supplement 2A–D*). By contrast, in mars1-2, the mutagenic cassette mapped to an intergenic of chromosome seven and tetrad analysis of WT x mars1-2 backcrosses (showing perfect 2:2 segregation) revealed that the causative mutation was unlinked from the cassette insertion (*Figure 2E*, *Figure 2—figure supplement 3A–C*) yet due to a single Mendelian mutation (*Figure 2—figure supplement 3A–C*). Whole genome sequencing of pooled progeny revealed a 13 kb deletion at the end of chromosome XVI, encompassing MARS1 along with two adjacent genes (*Figure 2D*, *Figure 2—figure supplement 3D–E*), indicating that the cpUPR-silencing phenotype in mars1-2 also arose from a mutation in MARS1. To corroborate this conclusion, we picked two additional MARS1 loss-of-function alleles from a C. reinhardtii mutant library (*Li et al., 2019*). As predicted, these mutants —mars1-3 and mars1-4, carrying insertions in MARS1 intron 19 and exon 18, respectively (*Figure 2D*)— were defective in inducing Vipp2 upon exposure to HL (*Figure 2F*). Tetrad analyses confirmed that the insertional cassette used to generate this library co-segregated with the cpUPR silencing phenotype in mars1-3 and mars1-4 (*Figure 2—figure supplement 4A–C*).

Moreover, other conditions that disrupt chloroplast protein homeostasis —namely chloroplast translation inhibition by spectinomycin treatment and oxidative stress by hydrogen peroxide exposure (*Erickson et al., 2015*; *Bobik and Burch-Smith, 2015*; *Blaby et al., 2015*)— likewise failed to trigger the cpUPR in these *MARS1* mutants (*Figure 2—figure supplement 4D*), further supporting a causative link between mutations in *MARS1* and the cpUPR-silencing phenotype.

Reverse transcriptase PCR (RT-PCR) analyses further validated that the *MARS1* mutants were defective in the expression of *MARS1* mRNA. As expected, *MARS1* mRNA was absent in *mars1-2*, *mars1-3* and *mars1-4* (*Figure 2—figure supplement 3D–E*, *Figure 2—figure supplement 4E*). By contrast, we detected residual *MARS1* mRNA levels in *mars1-1* cells, suggesting that a strong reduction in *MARS1* gene expression is sufficient to impair activation of the cpUPR (*Figure 2—figure supplement 4E*).

The Phytozome-annotated (URL: https://phytozome.jgi.doe.gov/pz/portal.html) *MARS1* gene model specifies an unusually long 5' untranslated region (5' UTR) that spans the first five exons of *MARS1* and predicts that the start site of the *MARS1* open reading frame is in the middle of exon 5 (*Figure 3A*, ATG(*ii*)). However, in the same gene model, an alternative in-frame translation start-site can be found in exon 1 (*Figure 3A*, ATG(*i*)). Interestingly, the coding sequence starting from ATG(*ii*) would give rise to a Mars1 protein with a potential N-terminal chloroplast target peptide, while the 138aa N-terminal extension translated from the alternative start codon ATG(*i*) predicts a Mars1 protein with a cytosolic localization. For complementation analyses, we generated two epitope-tagged *MARS1* transgenes (*MARS1-A* and *MARS1-D*). These two transgenes include the endogenous promoter, 5'UTR and 3'UTR, but the 3x-Flag epitope is in different positions (*Figure 3A*). In *MARS1-A*, we placed the 3x-Flag far downstream of the two putative translation start sites, yet upstream of the putative kinase domain. In *MARS1-D*, we placed the 3x-Flag epitope directly after ATG(*ii*), at the beginning of the putative N-terminal chloroplast target peptide that might be translated from this potential start site. Immunoblot and qPCR analyses of Vipp2 and Hsp22E/F confirmed that expression of either transgene could rescue the cpUPR-silencing phenotype of the *MARS1* mutants (*Figure 3A*, *Figure 3—figure supplement 1A–E* and *Figure 4D*). Moreover, the Mars1 protein could be detected by Flag immunoblot analysis not only in the case of *MARS1-A* but also when the *MARS1-D* transgene was used (*Figure 3A* and *Figure 4D*). Since the N-terminus of chloroplast stromal proteins is cleaved and promptly degraded upon organellar import, this result suggested that Mars1 was translated from ATG(*i*) and, therefore, would likely be localized in the cytosol. To support this notion, we performed biochemical fractionation and dual immunofluorescence microscopy using the internally tagged *MARS1* transgene (*MARS1-A*) (*Figure 3C–D*, *Figure 3—figure supplement 2A–B*). These orthogonal methods confirmed that Mars1 was enriched in the cytosol while depleted from the chloroplast, nuclear and mitochondrial compartments (*Figure 3C–D*).

To explore whether the predicted kinase activity of Mars1 would be involved in cpUPR signaling, we introduced a point mutation, which disrupts the conserved catalytic triad of the kinase (D1871A) in both *MARS1-A* and *MARS1-D* transgenes and tested the ability of these Flag-tagged *MARS1-KD* constructs (KD for k̲inase-d̲ead) to rescue *mars1* cells. In contrast to *MARS1*, expression of either *MARS1-A KD* or *MARS1-D KD* transgene failed to restore cpUPR signaling, as demonstrated by its lack of *VIPP2* and other cpUPR target gene induction upon Clp repression and HL stress (*Figure 3B* and *Figure 3—figure supplement 1A–D*). These results strongly suggest that an enzymatically active kinase domain in Mars1 is required for cpUPR signaling.

The discovery of a critical player in the cpUPR gave us a unique opportunity to examine the physiological role of the cpUPR during conditions of chloroplast proteotoxicity. To this end, we compared the sensitivity of WT and *mars1* cells to HL stress. Upon prolonged exposure to HL, *mars1* cells exhibited accelerated photobleaching and slower growth recovery relative to WT cells (*Figure 4A–B*, *Figure 4—figure supplement 1A*, *Figure 4—figure supplement 1—source data 1*). Notably, both phenotypes were rescued by expression of wild-type *MARS1* but not by *MARS1-KD* (*Figure 4A–B*, *Figure 4—figure supplement 1A*, *Figure 4—figure supplement 1—source data 1*).

We next tested the sensitivity of *MARS1* mutants towards metronidazole, a drug that selectively generates hydrogen peroxide in chloroplasts when cells engage in photosynthesis upon light exposure (*Schmidt et al., 1977*; *Dent et al., 2015*). *mars1* cells proved remarkably more sensitive than WT to metronidazole in photoheterotrophic and in phototrophic conditions, while their growth was not affected when grown in the dark (*Figure 4C*, *Figure 4—figure supplement 1B–F*). By contrast, WT and *mars1* cells were equally sensitive to tunicamycin, a chemical inducer of proteotoxic stress in

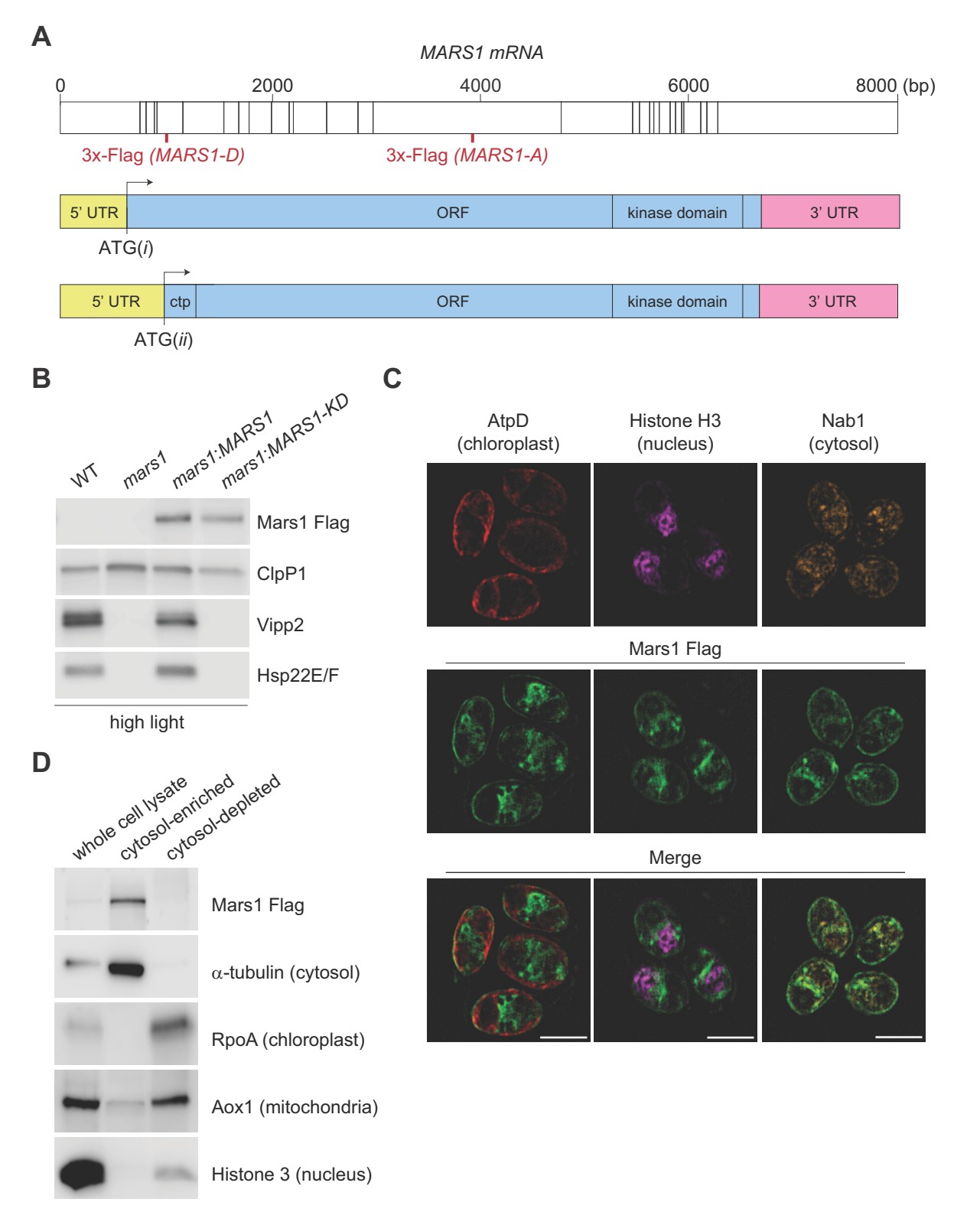

**Figure 3.** Characterization of Mars1. (**A**) In the upper diagram, the length of *MARS1* transcript is shown, the position of exon-exon boundaries is indicated with vertical lines while the different locations of the triple Flag epitope engineered in *MARS1-A* and *MARS1-D* transgenes are highlighted in red. In the lower diagrams, alternative models for the 5' untranslated region (5' UTR) (yellow) and open reading frame (ORF) (light blue) in *MARS1* transcript are shown as predicted for translation starts ATG(*i*) and ATG(*ii*). The N-terminal chloroplast transit peptide (ctp) is predicted by ChloroP

*Figure 3 continued on next page*

*Figure 3 continued*

(*Emanuelsson et al., 1999*) only if ATG(*ii*) is used as the translation start site. In both models, the position of the 3′ UTR (pink) and the kinase domain is the same. (**B**) Immunoblot analysis of samples prepared after HL treatment using antibodies against Flag for Mars1 detection, Vipp2, Hsp22E/F, and ClpP1 (loading control). Strains analyzed: *mars1* = *mars1-3*; *mars1:MARS1-A* = *mars1-3* transformed with a *MARS1-A* transgene containing a 3x-Flag epitope after Arg1167 of Mars1 (as shown in *Figure 3A*); *mars1:MARS1-A KD* = *mars1-3* transformed with a catalytically-inactive *MARS1-A* transgene bearing the kinase active site D1871A mutation. (**C**) Representative dual immunofluorescence images obtained by structured illumination microscopy of *mars1-3:MARS1-A* cells. Mars1 was detected with anti-Flag. Anti-AtpD, anti-Histone H3 and anti-Nab1 staining served as controls for the localization of the chloroplast, nucleus and cytosol, respectively. Scale bar: 5 μm. For imaging conditions and negative controls, see Supplementary Materials and *Figure 3—figure supplement 2*. (**D**) Immunoblot analysis of lysates fractionated by differential centrifugation from *mars1-3:MARS1-A* cells probed with the indicated antibodies against known markers of the cytosol (α-tubulin), chloroplast (RpoA, α-subunit of chloroplast RNA polymerase), nucleus (Histone H3) and mitochondria (Aox1, alternative oxidase 1).

The online version of this article includes the following figure supplement(s) for figure 3:

**Figure supplement 1.** A catalytic active Mars1 kinase is required for signaling during the cpUPR.
**Figure supplement 2.** Specific immunodetection of the Mars1 Flag protein.

the ER (*Figure 4—figure supplement 2A–C*) (*Yamaoka et al., 2018*). Moreover, as we observed for other conditions inducing the cpUPR, the metronidazole-mediated activation of cpUPR target genes, such as *VIPP2* and *HSP22E/F*, was dependent on Mars1 and its kinase activity (*Figure 4D*). Notably, by performing Flag affinity purification followed by mass spectrometry using Mars1-D and Mars1-D KD subjected to metronidazole treatment, we identified three serine residues (S69, S280 and S1888) that were selectively phosphorylated in Mars1-D but not in Mars1-D KD (*Supplementary file 1*). Furthermore, we detected peptide spectra derived from the first 138 aa of Mars1. Thus, taken together, our results strongly suggest that Mars1 is a cytosolic kinase that is required for the tolerance of chloroplast proteotoxic stress.

To characterize more comprehensively the function of Mars1 in transcriptional activation of cpUPR target genes, we compared the transcriptome of WT versus *mars1-1* cells by RNA sequencing following Clp repression and HL exposure (*Figure 5A–D*, *Figure 5—figure supplement 1A*). Given the previously reported complexity of cell stress responses in algae and higher plants (*Chan et al., 2016*; *Erickson et al., 2015*; *Bobik and Burch-Smith, 2015*), we aimed to identify a core set of *MARS1*-responsive genes under both cpUPR-inductive conditions (*Figure 5A–B*). Seven of the eight genes annotated as most highly co-expressed with *VIPP2* in the Phytozome database were up-regulated only in wild-type but not in *mars1* cells, including those encoding chloroplast small heat shock proteins (*Figure 5—figure supplement 1B*, *Figure 5—figure supplement 2—source data 1*). Similarly, the transcriptional activation of *CLPB3*, *DEG11* and stromal *APX,* which are evolutionarily conserved genes involved in chloroplast protein quality control and detoxification of reactive oxygen species, was impaired in *mars1* cells (*Figure 5A*). The *MARS1*-dependent transcriptome also comprised gene clusters involved in RNA metabolism, autophagy, and sulfur uptake (*Figure 5—figure supplement 1C*, *Figure 5—figure supplement 3A–B*). Importantly, *mars1* cells did not show reduced growth when subjected to sulfur deprivation, a different stress condition in which activation of autophagy and sulfur starvation genes are essential for cell survival (*Figure 5—figure supplement 4A–E*) (*Kajikawa et al., 2019*; *Zhang et al., 2004*). Thus, Mars1 selectively responds to chloroplast proteotoxic stress (*Ramundo and Rochaix, 2014*; *Heredia-Martínez et al., 2018*). Intriguingly, although *mars1* cells displayed a growth defect under HL conditions, several HL-controlled genes encoding components of the photosynthesis machinery did not require *MARS1* (*Figure 5C–D*, *Figure 5—figure supplement 5*). Likewise, the activation of genes that function in non-photochemical quenching (NPQ), a pathway that contributes to HL tolerance by dissipating excess energy as heat, was not impaired in *mars1* cells (*Erickson et al., 2015*; *Correa-Galvis et al., 2016*) (*Figure 5C–D*, *Figure 5—figure supplement 5*). Thus, the cpUPR entails a unique transcriptional response that is likely to act in concert with other known HL-tolerance mechanisms.

The publicly available transcriptomics data on Chlamydomonas circadian cell cycle show that *MARS1* belongs to a gene cluster that exhibits its expression peak at dawn and its expression trough at night (*Figure 5—figure supplement 2A*). The steady increase in *MARS1* mRNA level immediately preceding exposure to light supports the notion that the Mars1 kinase may be important for responding to light fluctuations. Indeed, in our transcriptomics dataset, *MARS1* mRNA was expressed roughly 10-fold and 2-fold more upon HL exposure and ClpP1 repression,

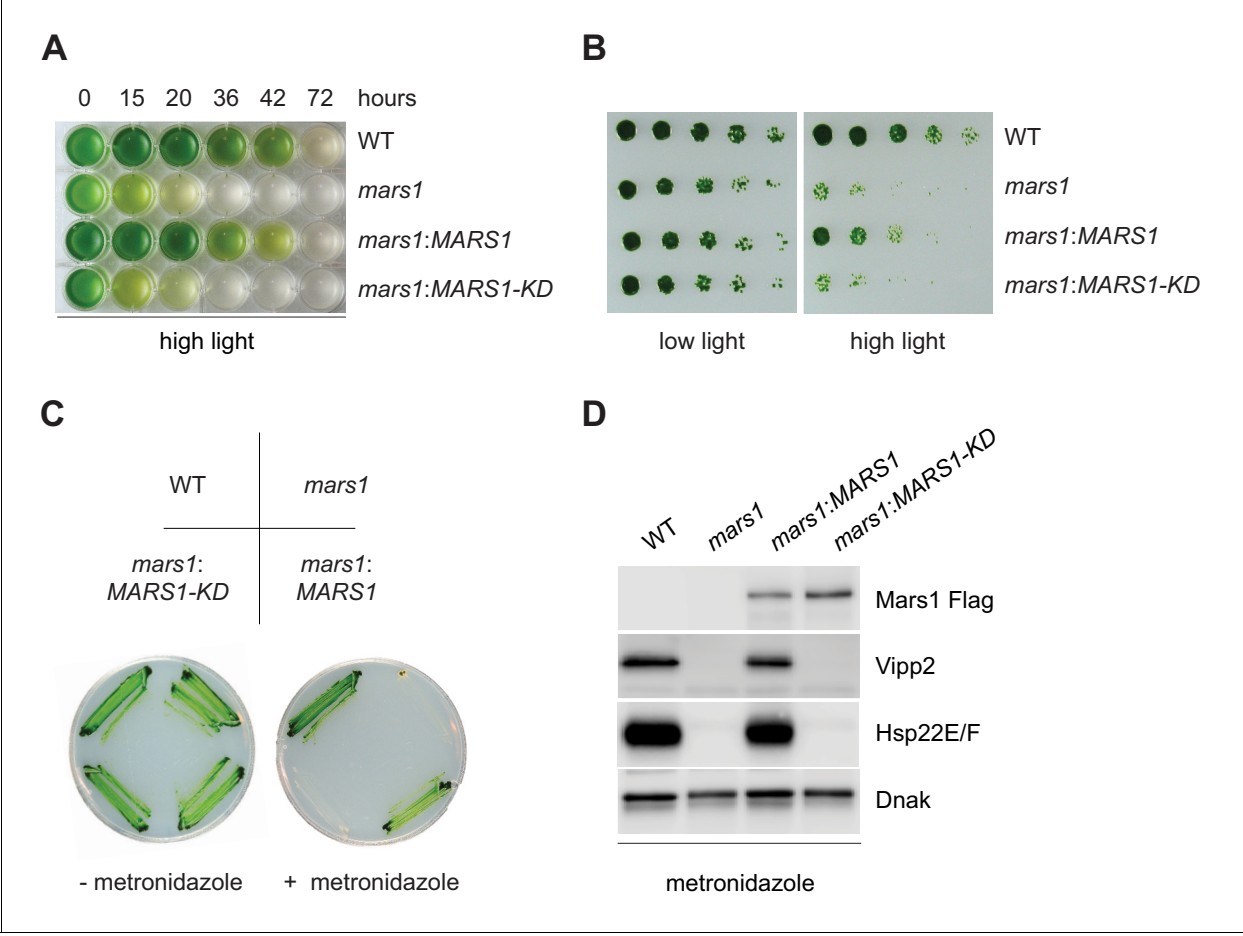

**Figure 4.** *mars1* cells are hypersensitive to photooxidative stress. (**A**) Liquid TAP cultures of WT, *mars1*, *mars1:MARS1-D* and *mars1:MARS1-D KD* at different time points upon HL exposure. Alleles used: *mars1* = *mars1-3*; *mars1:MARS1-D* = *mars1-3* transformed with the *MARS1-D* transgene containing a 3x-Flag epitope after Met139; *mars1:MARS1-D KD* = *mars1-3* transformed with a catalytically-inactive *MARS1-D* bearing the kinase active site D1871A mutation. (**B**) Strains described in *Figure 4A* were spotted onto TAP agar plate in 4-fold serial dilutions before and after exposure to high light for 27 hr. Photographs of untreated and treated cells were taken after 6 and 7 days, respectively, of growth in low light. (**C**) Strains described in *Figure 4A* were streaked on -/+ 1.5 mM metronidazole TAP agar plates. Photographs were taken after 4 days of growth in normal light. (**D**) Immunoblot analysis of samples prepared from strains described in *Figure 4A* treated with 1.1 mM metronidazole for 15 hours. Detection with antibodies against Flag (Mars1), Vipp2, Hsp22E/F, and ClpP1 (loading control).

The online version of this article includes the following source data and figure supplement(s) for figure 4:

**Figure supplement 1.** Mars1 confers protection against photooxidative stress.

**Figure supplement 1—source data 1.** Chlorophyll measurements during high light stress.

**Figure supplement 2.** *mars1* cells can cope with ER stress.

respectively (*Figure 5—figure supplement 1B*, *Figure 5—figure supplement 2B*). However, in the hypomorphic *mars1-1* mutant allele, where residual amounts of *MARS1* transcript were detected, (*Figure 5—figure supplement 1B*, *Figure 5—figure supplement 2B*), the stress-dependent *MARS1* upregulation was lost. It is therefore possible that the activation of a mild chloroplast UPR may be part of the physiological circadian cycle and that *MARS1* gene expression may be regulated through a positive feedback loop.

Finally, we took advantage of the serendipitous finding that expression of the *MARS1-E* transgene (*Figure 6A*), bearing a 6x-Flag tag insertion after Leu402 of Mars1, upregulated both Vipp2 and Hsp22E/F even in the absence of stress (*Figure 6B–C*). The dominant activating pheno-type of *MARS1-E* was not observed in wild-type cells expressing other Flag-tagged alleles of *MARS1* (*Figure 6—figure supplement 1A*), and was dependent on Mars1's enzymatic function, as it was blocked by the D1871A kinase inactivating mutation (*Figure 6B–C*, *Figure 6—figure supplement*

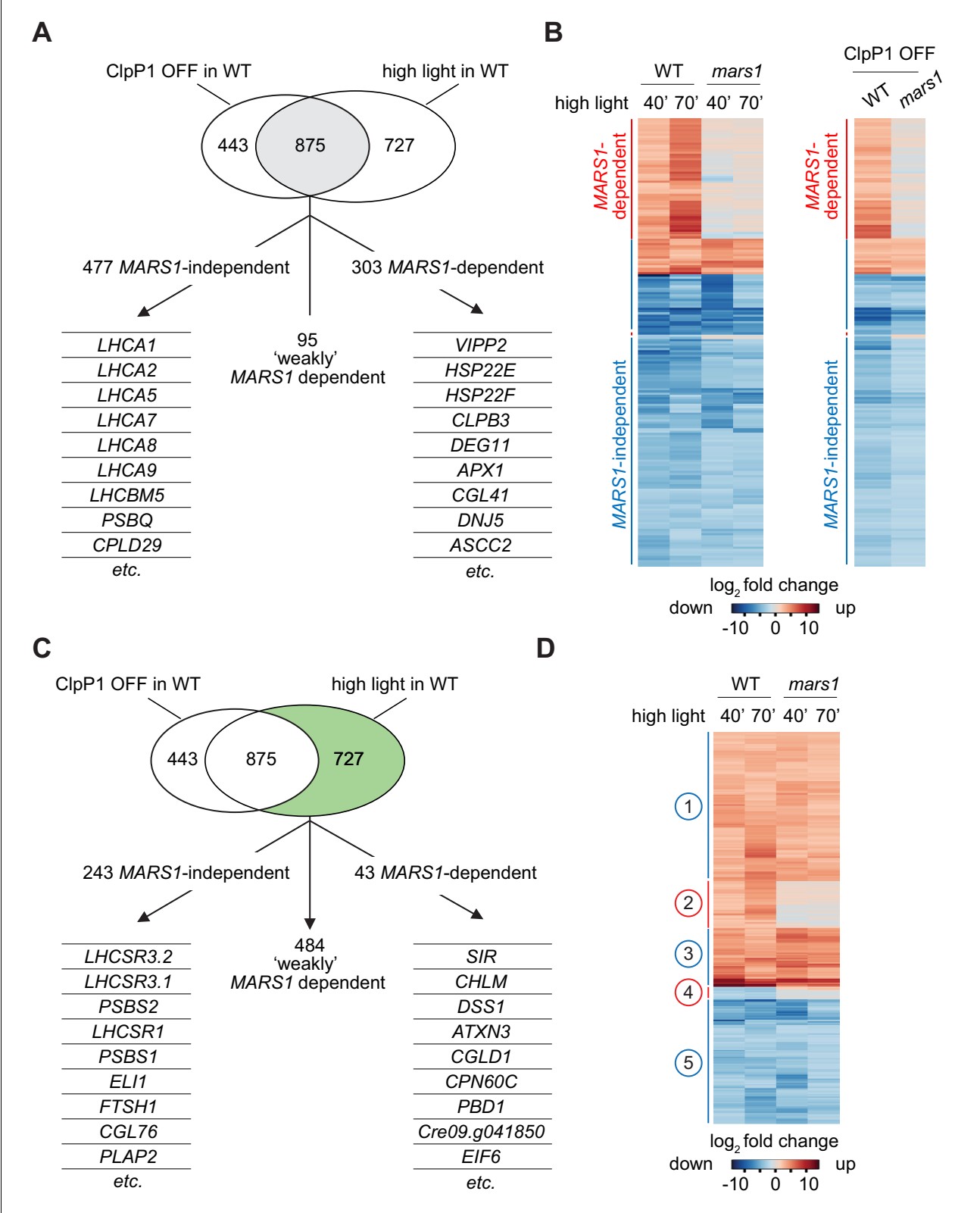

**Figure 5.** *mars1* cells do not activate the cpUPR transcriptional program. (**A**) Venn diagram highlighting the transcriptional changes elicited genetically (ClpP1 repression) or physiologically (HL exposure) in Chlamydomonas cells , as determined by RNA sequencing (overlap in gray). The 875 common stress-responsive genes are defined as genes whose expression showed at least a 4-fold change upon stress (p<0.001) and were consistently up- or down-regulated in both high light and ClpP1 repression. These genes are further subdivided in three groups: 1) *MARS1*-dependent, being

*Figure 5 continued on next page*

*Figure 5 continued*

unresponsive (<2-fold change) when *MARS1* was disrupted; 2) 'weakly' *MARS1* dependent, being only mildly induced or inhibited (<4-fold but >2-fold) upon stress in the mars1 background; and 3) *MARS1*-independent, being still responsive (<2-fold change) in absence of *MARS1* expression. A short list of genes belonging to each category (*MARS1*-dependent and *MARS1*-independent) is provided. Full lists of *MARS1*-dependent and *MARS1*-independent genes are available through Figshare (https://figshare.com/s/992706a610ce6b71f03c and https://figshare.com/s/66417c2b28f3110b8077). (B) Heat-map comparing gene expression patterns of the 875 common stress-responsive genes (as defined in *Figure 5A*) in WT and *mars1* cells upon exposure to HL (for 40 or 70 min) or in ClpP1-nonpermissive conditions (+Vit). (C) The same Venn diagram shown in *Figure 5A* highlighting (in green) genes that are preferentially responsive to HL exposure (>4-fold change only upon HL stress, p<0.001, and consistently up- or down-regulated in both time points during HL stress). These 727 genes are further categorized based on their gene expression dependency on *MARS1*. A short list of genes belonging to each category (*MARS1*-dependent and *MARS1*-independent) is provided. *MARS1*-dependent genes are related to chloroplast protein folding and degradation, protein translation or are poorly characterized. By contrast, the list of *MARS1*-independent genes includes key regulators of nonphotochemical quenching such as *LHCSR* and *PSBS* genes. HL = high light; ClpP1 OFF = ClpP1 repression. (D) Heatmap showing genes that are preferentially responsive to HL exposure. Five clusters of genes are highlighted on the side. Numbers circled in blue and red indicate gene clusters not affected and affected by *MARS1* disruption, respectively.

The online version of this article includes the following source data and figure supplement(s) for figure 5:

**Figure supplement 1.** *mars1* cells do not induce the cpUPR transcriptional program.
**Figure supplement 2.** *MARS1* gene expression pattern.
**Figure supplement 2—source data 1.** *MARS1* transcript levels (RPKM values).
**Figure supplement 3.** *mars1* cells do not activate autophagy and sulfur starvation genes during cpUPR inducing conditions.
**Figure supplement 4.** *mars1* cells activate sulfur starvation genes and survive in sulfur-limiting conditions.
**Figure supplement 5.** Regulation of photosynthesis-associated genes is not affected in *mars1* cells.

*1B*). We consider it likely that the constitutive cpUPR phenotype results from a fortuitous activation of Mars1, perhaps by inactivation of an auto-repressive feature of the enzyme or by modification of a protein-protein or a protein-metabolite interface. *MARS1-E* was sufficient to trigger activation of the canonical cpUPR, as confirmed by qPCR analysis of *MARS1*-dependent transcripts, such as those in the *VIPP2* co-expression cluster (*Figure 6C*). Conversely, *MARS1*-independent transcripts such as those involved in NPQ were unaffected by expression of *MARS1-E* (*Figure 6C*). These results suggest that Mars1 is an integral component directly involved in cpUPR signaling since its activation is not only required but can also be sufficient to induce the cpUPR transcriptional program. Notably, cells expressing *MARS1-E* exhibited a higher resistance to metronidazole and to HL stress (*Figure 6D–E*), indicating that induction of the cpUPR can confer a growth advantage in the presence of chloroplast proteotoxicity.

## Conclusions

In summary, our results suggest that, upon onset of chloroplast proteotoxic stress, a signal transduction pathway, originating in the chloroplast, leads to activation of the cytosolic kinase Mars1, which in turn orchestrates the cpUPR transcriptional program. Activation of cpUPR through Mars1 mitigates photooxidative stress and delays photobleaching. However, loss of Mars1 does not impair expression of genes involved in non-photochemical quenching. Thus, the exact mechanism by which the cpUPR pathway confers photoprotection remains to be deciphered. Input conditions that activate the cpUPR, as well as cpUPR target genes identified to date, are phylogenetically conserved from *C. reinhardtii* to *A. thaliana* (*Llamas et al., 2017*; *Zybailov et al., 2009*; *D'Andrea et al., 2018*). Thus, despite not yet having identified a functional ortholog of *MARS1* in higher plants, it is reasonable to assume that the cpUPR's previously unknown role in protecting cells against photooxidative stress would likewise be conserved. This notion is particularly appealing in light of the observation that basal induction of the cpUPR conferred a protective effect in response to stress and that, conversely, loss of Mars1 activity profoundly sensitized cells towards HL and other chloroplast stressors. Hence, engineering plants with constitutive cpUPR activation may be a promising strategy to enhance their tolerance to environmental stresses.

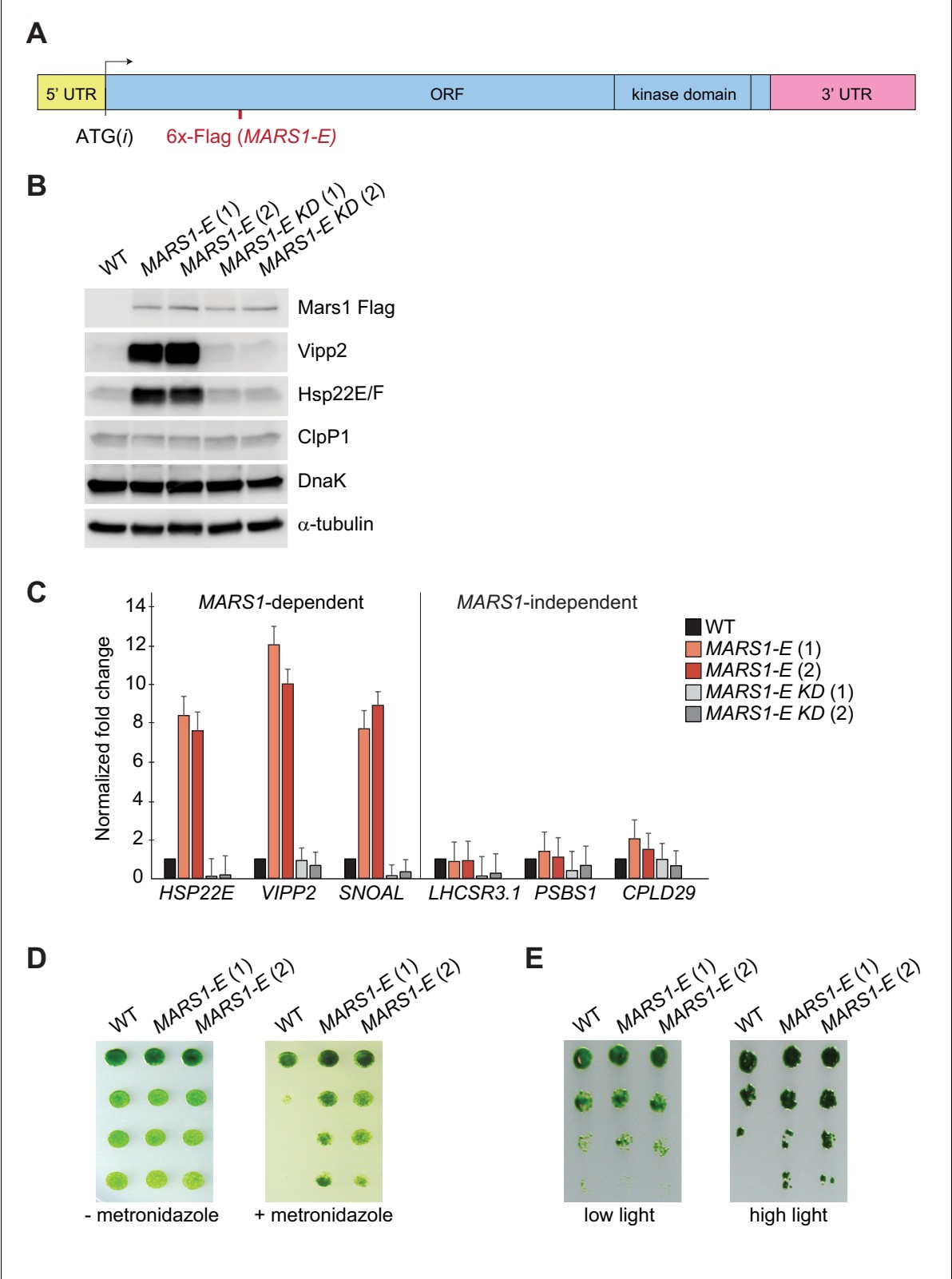

**Figure 6.** Basal induction of the cpUPR renders cells more resistant to chloroplast stress. (**A**) A diagram of *MARS1-E* transcript showing the position of the 6x-Flag epitope inserted after Leu402 of Mars1. (**B**) Immunoblot analysis of samples prepared from cells grown under normal conditions, using antibodies described in *Figure 3*. Strains used: *MARS1-E* (1) and (2) = WT cells expressing a *MARS1-E* transgene; *MARS1-E KD* (1) and (2) = WT cells expressing a catalytically-inactive *MARS1-E* transgene bearing the D1871A mutation. (**C**) Expression level of *MARS1*-dependent or *MARS1*-independent

*Figure 6 continued on next page*

*Figure 6 continued*

transcripts determined by quantitative PCR under normal growth conditions in cpUPR constitutive-active cells described in *Figure 6B*. Reference gene for normalization: *GBLP*. (**D**) cpUPR constitutive-active cells (described in *Figure 6B*) were grown in liquid TAP until logarithmic phase, diluted to the same cell count and spotted onto +/- 2.2 mM metronidazole agar plates using 1.5-fold dilutions between spots. Photographs of untreated and treated cells were taken after 3 and 6 days, respectively, of growth in normal light. (**E**) Cells as in *Figure 6D* spotted onto TAP agar using 4-fold serial dilutions before or after exposure to HL. Photographs were taken after 7 days of growth in normal light.

The online version of this article includes the following figure supplement(s) for figure 6:

**Figure supplement 1.** The *MARS1-E* transgene causes mild induction of cpUPR signaling.

# Materials and methods

## Key resources table

| Reagent type (species) or resource | Designation | Source or reference | Identifiers | Additional information |
|---|---|---|---|---|
| Gene (*C. reinhardtii*) | *clpP1* | GenBank | L28803.1 | |
| Gene (*C. reinhardtii*) | *VIPP2* | Phytozome | Cre11.g468050 | |
| Gene (*C. reinhardtii*) | *MARS1* | Phytozome | Cre16.g692228 | |
| Recombinant DNA reagent | YFP 3x-FLAG | GenBank | ANF29833.1 | |
| Recombinant DNA reagent | pMJ016c | GenBank | KX077951.1 | |
| Antibody | anti-ClpP1 | gift from Francis-André Wollman and Olivier Vallon | | 1:5000 |
| Antibody | anti-Vipp2 (rabbit, polyclonal) | developed during this study | | (1:3000) |
| Antibody | anti-FLAG (mouse, monoclonal) | Sigma | F1804 | (1:3000 for IB; 1:500 for IF) |
| Antibody | anti-Hsp22E/F (rabbit, polyclonal) | gift from Michael Schroda | | (1:10000) |
| Antibody | anti-DnaK (rabbit, polyclonal) | gift from Jean David Rochaix | | (1:10000) |
| Antibody | anti-α tubulin (rabbit, polyclonal) | Sigma | T5168 | (1:10000) |
| Antibody | anti-Histone H3 (rabbit, polyclonal) | Agrisera | AS10 710 | (1:10000 for IB; 1:500 for IF) |
| Antibody | anti-AtpD (rabbit, polyclonal) | Agrisera | AS10 1590 | (1:500) |
| Antibody | anti-Nab1 (rabbit, polyclonal) | Agrisera | AS08 333 | (1:500) |
| Antibody | anti-AtpD (rabbit, polyclonal) | Agrisera | AS10 1590 | (1:500) |
| Antibody | anti-RpoA (rabbit, polyclonal) | gift from Jean David Rochaix | | (1:10000) |
| Antibody | anti-holo Rubisco (rabbit, polyclonal) | gift from Jean David Rochaix | | (1:10000) |

*Continued on next page*

*Continued*

| Reagent type (species) or resource | Designation | Source or reference | Identifiers | Additional information |
|---|---|---|---|---|
| Antibody | anti-Aox1 (rabbit, polyclonal) | Agrisera | AS06 152 | (1:2000) |
| Antibody | anti-Sultr2 (rabbit, polyclonal) | gift from Arthur Grossman | | (1:3000) |
| Antibody | anti-Hsp90 (rabbit, polyclonal) | Agrisera | AS06 174 | (1:10000) |
| Commercial assay or kit | KOD Hot Start DNA Polymerase | ThermoFisher Scientific | 71086–3 | |
| Commercial assay or kit | Phusion High-Fidelity DNA Polymerase | ThermoFisher Scientific | F530L | |
| Commercial assay or kit | PureYield Plasmid Miniprep System | Promega | A1222 | |
| Commercial assay or kit | NucleoSpin Gel and PCR Clean-up | Clontech | 740609 | |
| Commercial assay or kit | Direct-zol RNA Miniprep Plus | Zymo Research | R2070 | |
| Commercial assay or kit | Alexa Fluor 488 Tyramide SuperBoost Kit, goat anti-mouse IgG | ThermoFisher Scientific | B40941 | |
| Commercial assay or kit | In-Fusion HD cloning plus | Takara | 638910 | |
| Commercial assay or kit | CircLigase II ssDNA | Epicentre | CL9025K | |
| Commercial assay or kit | PrimeScript 1 st strand cDNA Synthesis Kit | Takara | 6110A | |
| Commercial assay or kit | iQ SYBR Green Supermix - Bio-Rad | Bio-Rad | 170–8880 | |
| Commercial assay or kit | Dynabeads kilobase Binder Kit | Invitrogen | 60101 | |
| Commercial assay or kit | Kapa mRNA HyperPrep kit | Roche | KK8540 | |
| Commercial assay or kit | PrepX DNA Library Kit (400075) | Takara | 640101 | |
| Commercial assay or kit | SuperSignal West Femto | ThermoFisher Scientific | 34095 | |
| Commercial assay or kit | SuperSignal West Dura | ThermoFisher Scientific | 34075 | |
| Chemical compound, drug | N6,2'-O-Dibutyryladenosine 3',5'-cyclic monophosphate sodium salt | Sigma | D0627 | |
| Chemical compound, drug | Thiamine hydrochloride | Sigma | T4625 | |
| Chemical compound, drug | Vitamin $B_{12}$ | Sigma | V6629 | |
| Chemical compound, drug | digitonin, high purity | Calbiochem | 300410 | |
| Chemical compound, drug | Metronidazole | Sigma | M3761 | |
| Software, algorithm | Sequence Data Analysis | Geneious | | |
| Software, algorithm | Sequence Data Analysis | Snapgene | | |

*Continued on next page*

*Continued*

| Reagent type (species) or resource | Designation | Source or reference | Identifiers | Additional information |
|---|---|---|---|---|
| Software, algorithm | Microscopy imaging interface | Zeiss | ZEN | |
| Software, algorithm | Image processing | NIH | ImageJ | |
| Other | electroporation system | Biorad | Gene Pulser II | |
| Other | electroporation system | Nepagene | Nepa21 | |
| Other | colony picker | Norgren Systems | CP7200 | |
| Other | colony manipulation robot | Singer | Rotor HAD | |
| Other | Fluorescence scanner | GE Healthcare | Typhoon Trio | |
| Other | Protran Nitrocellulose Hybridization Transfer Membrane | Perkin Elmer | NBA083C001EA | |
| Other | Imaging system | LI-COR Biosciences | Odyssey CLx | |
| Other | Plant LED Grow Light | Phlizon 2017 | 2000W | |
| Other | Smart Sensor | Wifi Gateway | G1 | |
| Other | Smart Sensor | Sensor Push | HT1 | |

## General maintenance of *C. reinhardtii* strains

All *C. reinhardtii* strains were maintained on Tris-Acetate-Phosphate (TAP) solid media (1.6% agar, USP grade, Thermo Fisher Scientific) with revised Hutner's trace elements (*Kropat et al., 2011*) at 22°C in low light (~10–20 µmol photons m$^{-2}$ s$^{-1}$). Lines harboring the ClpP1 repressible gene were maintained in the media supplemented with 100 µg ml$^{-1}$ spectinomycin (Sigma). Lines harboring a mutagenic cassette disrupting the *MARS1* gene were maintained in the media supplemented with 20 µg ml$^{-1}$ paromomycin (Sigma). Lines harboring a *MARS1* transgene construct were maintained in the same conditions with solid media supplemented with 20 µg ml$^{-1}$ hygromycin (Thermo Fisher Scientific). Sulfur-depleted TAP liquid and agar pates were prepared as previously described (*Davies et al., 1994*). Generally, during liquid growth, no antibiotic was supplemented. For the experiments shown in *Figure 4—figure supplement 2A–C*, TAP liquid cultures and agar plates were supplemented with 5 µg/ml or 0.2 µg/ml tunicamycin (EMD Millipore), respectively. All strains used in this study are listed in *Table 1*.

## Generation of the cpUPR reporter strain

The cpUPR reporter strain (CrPW1) was generated by nuclear transformation of the *C. reinhardtii* ClpP1 repressible strain (DCH16; *Ramundo et al., 2014*), using 300 ng of Nde1-linearized pPW3217 plasmid *Table 2* for plasmids used in this study). Nde1 and all the other restriction enzymes described in this publication were purchased from NEB. The nuclear transformation was carried out via electroporation as described below (section on 'Insertional mutagenesis'). Transformants isolated on TAP agar plates containing 20 µg ml$^{-1}$ hygromycin and tested by Flag immunoblot analysis upon ClpP1 repression and exposure to HL. As previously observed (*Ramundo et al., 2014*; *Ramundo et al., 2013*), during random insertion of a construct with regulatory regions in its promoter, less than 10% of the hygromycin resistant transformants preserved the correct gene expression pattern of the downstream coding sequence. Among them, we selected CrPW1 for further studies. The pPW3217 plasmid, containing a minimum region of the *VIPP2* gene promoter, its 5' untranslated region and its first exonexon intrintrone yellow fluorescent protein coding sequence (YFP CDS), C-terminally appended to a triple Flag epitepitope the 3' untranslated region of the *RBCS2* gene, was generated by In-Fusion cloning (Clontech). The *VIPP2* genomic fragment was amplified from genogenomic with primers SR510 and SR502 (see *Table 3* for primers used in this study). The YFP CDS fused to a 3x-Flag epitope was amplified from pLM005 (Chlamydomonas Resource Center) with primers SR503 and SR504. The 3' untranslated region of the *RBCS2* gene was amplified from pHyg3 (*Berthold et al., 2002*) with primers SR505 and SR506. All PCRsPCRse

**Table 1.** Strains table.

| CrPW number | Strain name | Short description | Reference |
|---|---|---|---|
| | CC-4533 | parental strains used to generate *mars1-3* and *mars1-4* | (*Li et al., 2016*) |
| | CC-124 | wild-type used for *mars1-1* and *mars1-2* genetic backcrosses | (available at the Chlamydomonas Resource Center) |
| | A31 | parental strain of DCH16 | (*Ramundo et al., 2013*) |
| | DCH16 | ClpP1 repressible strain | (*Ramundo et al., 2013*) |
| CrPW1 | A1N5 | cpUPR reporter strain | (generated during this study) |
| CrPW2 | ACT C6 | YFP positive strain | (generated during this study) |
| CrPW3 | DRB1 | YFP positive strain | (generated during this study) |
| CrPW4 | *mars1-1* | Cre16.g692228 mutant allele | (generated during this study) |
| CrPW5 | *mars1-2* | Cre16.g692228 mutant allele* -Full genotype described below | (generated during this study) |
| CrPW6 | *mars1-3* | (available at the Chlamydomonas Resource Center) | Cre16.g692228 mutant allele Clip ID: LMJ.RY0402.195536 |
| CrPW7 | *mars1-4* | (available at the Chlamydomonas Resource Center) | Cre16.g692228 mutant allele Clip ID: LMJ.RY0402.189144 |
| CrPW8 | E12 | wild-type like progeny from backcross of mars1-1 to CC124 (used for RNA-seq analysis) | (generated during this study) |
| CrPW9 | F2 | *MARS1* mutant progeny from backcross of *mars1-1* to CC124 (used for RNA-seq analysis) | (generated during this study) |
| CrPW10 | D2C4 | wild-type like progeny from backcross of *mars1-2* to CC124 (used for complementation analysis)† | (generated during this study) |
| CrPW11 | D2C3 | *MARS1* mutant progeny from backcross of mars1-2 to CC124 (used for complementation analysis)† | (generated during this study) |
| CrPW12 | M22 | *mars1-3:MARS1-A* strain | (generated during this study) |
| CrPW15 | FMW14 | *mars1-3:MARS1-D* strain | (generated during this study) |
| CrPW16 | KDM14 | *mars1-3:MARS1-A KD* strain | (generated during this study) |
| CrPW17 | FMD17 | *mars1-3:MARS1-D KD* strain | (generated during this study) |
| CrPW18 | 189 N25 | *mars1-1:MARS1-A* strain | (generated during this study) |
| CrPW19 | FKD7 | *mars1-1:MARS1-D* strain | (generated during this study) |
| CrPW20 | pKP29 B30 | *mars1-1:MARS1-D KD* strain | (generated during this study) |
| CrPW21 | pKP30 D7 | *mars1-1:MARS1-D KD* strain | (generated during this study) |
| CrPW22 | DCM2 | *mars1-2:MARS1-A* strain | (generated during this study) |
| CrPW23 | DCM5 | *mars1-2:MARS1-A* strain | (generated during this study) |
| CrPW24 | DCM10 | *mars1-2:MARS1-A* strain | (generated during this study) |
| CrPW25 | DCM19 | *mars1-2:MARS1-A* strain | (generated during this study) |
| CrPW26 | DCM21 | *mars1-2:MARS1-A* strain | (generated during this study) |
| CrPW27 | W153 | CC-4533 transformed with *MARS1-E* transgene, strain (a) | (generated during this study) |
| CrPW28 | W155 | CC-4533 transformed with *MARS1-E* transgene, strain (b) | (generated during this study) |
| CrPW29 | WKD4 | CC-4533 transformed with *MARS1-E KD* transgene, strain (a) | (generated during this study) |

*Table 1 continued on next page*

*Table 1 continued*

| CrPW number | Strain name | Short description | Reference |
|---|---|---|---|
| CrPW30 | WKD16 | CC-4533 transformed with MARS1-E KD transgene, strain (a) | (generated during this study) |
| CrPW31 | WFM2 | CC-4533 transformed with MARS1-D transgene, strain (a) | (generated during this study) |
| CrPW43 | ire1 | Cre08.g371052 mutant allele Clip ID: LMJ.RY0402.122895 | (available at the Chlamydomonas Resource Center) |
| CrPW44 | snrk2 | Cre02.g075850 mutant allele Clip ID: LMJ.RY0402.187019 | (available at the Chlamydomonas Resource Center) |

*mars1-2 has a total of three mapped genomic disruptions.

[1]The chromosome 16 deletion which encompasses Cre16.g692228 (MARS1), Cre16.g692340, and Cre16.g692452.

[2]The full Paromomycin cassette was found in an intergenic region on chromosome 7, 958 bp downstream of gene Cre07.g336300. A portion of a gene-Cre02.g108450 (5'UTR-intron 4) was found directly upstream of this Paromomycin cassette (in intergenic region of Chromosome 7).

[3]The locus for the Cre02.g108450 gene itself has a deletion spanning the 5'UTR-intron 4.

[†]D2C3 and D2C4 were offspring isolated upon backcrossing mars1-2 to CC-124 three times. D2C4 contains the wild-type MARS1 gene whereas D2C3 contains the MARS1 deletion. In both strains, all other markers (Hygromycin, Paromomycin, and Spectinomycin) as well as the abovementioned Cre02.g108450 deletion were crossed out.

performed using Phusion Hotstart II polymerase (Thermo Fisher Scientific). PCR products were gel-extracted using NucleoSpin Gel and PCR Clean-Up Kit (Takara) according to manufacturer's instructions, and they were further purified by ultrapure phenol:chloroform:isoamyl alcohol (25:24:1, v/v/v) (Life Technologies) extraction and ice-cold ethanol (Sigma) precipitation. These three purified DNA fragments were then mixed with a purified and linearized pHyg3 vector, previously digested by PciI and EcoRV, and incubated with the In-Fusion reagents (Takara) as per manufacturer's instructions. The In-Fusion product was transformed in Stellar competent cells (Takara) according to manufacturer's instructions and putative positive clones were selected in LB solid media (1.7% agar) supplemented with ampicillin after overnight incubation. The resulting plasmid, pPW3217, was purified using the Kit PureYield Plasmid Miniprep System (Promega) and verified by analytical digestion and sequencing. All constructs made by In-Fusion cloning for this publication follow this same protocol from transformation through plasmid isolation.

## Preparation of the paromomycin cassette for insertional mutagenesis

The mutagenic DNA cassette was isolated by restriction enzyme digestion of pMJ016c (provided by the Jonikas laboratory), which contains the HSP70-RBCS2 chimeric promoter, the paromomycin resistance gene AphVIII, and the PSAD and RPL12 chimeric terminator (Mackinder et al., 2016). Using the Mly1 enzyme, a blunt fragment of 2204 bp (containing the mutagenic DNA cassette) was isolated and extracted from a 1% agarose gel through the NucleoSpin Gel and PCR Clean-Up Kit (Takara) per the manufacturer's instructions. To further remove possibly contaminating DNA, the mutagenic DNA cassette was electrophoresed on a 1% agarose gel, extracted and repurified as explained above.

## Insertional mutagenesis and maintenance of mutant library prior to the screen

A 1-liter liquid culture of cpUPR reporter strain (CrPW1) was grown in TAP medium in low light (~30 $\mu$mol photons m$^{-2}$ s$^{-1}$) to a density of about $2–4 \times 10^6$ cells ml$^{-1}$. Cells were collected at room temperature (RT) by centrifugation at 3000 x $g$ for 5 min and gently resuspended in TAP supplemented with 40 mM sucrose at $2 \times 10^8$ cells ml$^{-1}$. Multiple aliquots of 250 $\mu$L of cell suspension were then transferred into Gene Pulser electroporation cuvettes (0.4 cm gap, Bio-Rad) and incubated at 16°C for 5–30 min. In each cuvette, about 20 ng of mutagenic DNA cassette was added to the cell suspension and quickly mixed by pipetting. Electroporation was performed immediately using a Gene Pulser II electroporation system (Bio-Rad) with the following parameters: capacitance = 25 $\mu$F and

**Table 2.** Plasmids Table.

| Plasmid name (nickname/official name) | Used for | Reference |
|---|---|---|
| pLM005 | for amplification of the *YFP* coding sequence | (*Mackinder et al., 2016*) |
| pHyg3 | for amplification of the *RBCS2* 3'UTR sequence and cloning of the Hygromycin resistance cassette | (*Berthold et al., 2002*) |
| pMJ016c | for insertional mutagenesis | (*Li et al., 2016*) |
| pRAM118/pPW3216 | for gene tagging and subcloning | (*Li et al., 2019*) |
| pRAM103.5/pPW3217 | For generation of the cpUPR reporter strain | (generated during this study) |
| pRAM185.2/pPW3218 | For *MARS1* cloning (untagged *MARS1* transgene) | (generated during this study) |
| pRAM189 M2/pPW3219 | For *MARS1* cloning (*MARS1-A* transgene) | (generated during this study) |
| pKP29 /pPW3222 | For *MARS1* cloning (*MARS1-D* transgene) | (generated during this study) |
| pRAM184.1 /pPW3223 | For *MARS1* cloning (*MARS1-E* transgene) | (generated during this study) |
| pHT20.1/pPW3224 | For *MARS1* cloning (catalytically-dead *MARS1-A* transgene) | (generated during this study) |
| pKP30/pPW3225 | For *MARS1* cloning (catalytically-dead *MARS1-D* transgene) | (generated during this study) |
| pHT6/pPW3226 | For *MARS1* cloning (catalytically-dead *MARS1-E* transgene) | (generated during this study) |

voltage = 800 V. Electroporated cells from each cuvette were then diluted into 8 ml TAP supplemented with 40 mM sucrose and allowed to recover overnight by gentle agitation in very dim light (5–10 µmol photons $m^{-2}$ $s^{-1}$). The next day, cells were collected by centrifugation at 1000 x g for 5 min, resuspended in 1 ml of TAP medium, plated on TAP agar plates containing 20 µg $ml^{-1}$ paromomycin and incubated in darkness for about three weeks before picking colonies.

Approximately 55,000 total mutants were picked and re-arrayed in a 384-colony format on rectangular agar plates (Singer Instruments) using a Norgren CP7200 colony-picking robot. In each 384-mutant array plate, the last two rows were kept empty to include internal positive and negative controls for the next stage of the screen (for details, read section 'Execution of YFP mutant screen on agar plates'). This library of mutants (of approximately 150 agar plates) was grown in complete darkness at 22°C and propagated every 3–4 weeks by robotically passaging the mutant arrays to fresh 1.5% agar solidified TAP medium containing 100 µg $ml^{-1}$ spectinomycin using a Singer RoToR robot (Singer Instruments). Unfortunately, numerous mutants were lost during propagation due to a widespread contamination event.

## Execution of YFP mutant screen on agar plates

To screen for YFP silencing or activating mutants, rectangular agar plates, each containing 97 ml of TAP medium -/+ Vit (400 µM thiamine-HCl (Sigma) and 80 ng $ml^{-1}$ of vitamin $B_{12}$ (Sigma)), were prepared. In each agar plate (-/+ Vit), 12 colonies of the cpUPR reporter strain (CrPW1), 12 colonies of the parental ClpP1 repressible strain (DCH16) and 12 colonies of 2 different positive YFP expressor strains (CrPW2 and CrPW3) were robotically spotted in the last two rows of the 384-colony array to be used as internal positive and negative controls during the YFP screen (see scheme in section 'Semi-automated identification of YFP mutants through Image-J macroscripts'). Next, insertional mutants (freshly propagated) were spotted onto these same plates. Plates were incubated in dim light (20–30 µmol photons $m^{-2}$ $s^{-1}$) at 25°C and were imaged after 2 and 6 days using a Typhoon TRIO fluorescence scanner (GE Healthcare). For each round, 12 plates of insertional mutants (6 without and six with Vit) were simultaneously scanned with the settings described below. Chlorophyll

**Table 3.** Primers Table.

| Primer name | 5'–>3' sequence |
| --- | --- |
| oMJ598 | b-CAGGCCATGTGAGAGTTTGC<br>(b = biotinylated) |
| oMJ619 | /5Phos/AGATCGGAAGAGCGTCGTGTAGGG<br>AAAGAGTGTAGATCTCGGTGGTCGCCGTATC<br>ATTACTCAGTAGTTGTGCGATGGATTGATG/3ddc/<br>(/5Phos/=phosphorylated;/3ddc/=dideoxycytidine<br>(to prevent self-ligation) |
| oMJ621 | AATGATACGGCGACCACCGAGATCTACACTC<br>TTTCCCTACACGACGCTCTTCCGATCT |
| oMJ1234 | b-GCAGCCAAACCAGGATGATG (b = biotinylated) |
| oMJ1239 | aattaaccctcactaaagCAATCATGTCAAGCCTCAGC |
| T3_3'_oMJ016c 11/23 | aattaaccctcactaaagGGTCGAGCCTTCTGGCAGA |
| T3_5'_oMJ016c 11/24 | aattaaccctcactaaaggGCGGAGACGTGTTTCTGAC |
| SR502 | tgctcaccatACTAGTGAGCACGCTGCGA |
| SR503 | gctcactagtATGGTGAGCAAGGGCGAG |
| SR504 | gggatccttaagatctTTACTTGTCGTCATCGTCCTT |
| SR505 | cgacaagtaaagatctTAAGGATCCCCGCTCCGTG |
| SR506 | gcgcaagaaagaagcttgatatcCGCTTCAAATACGCCCAGC |
| SR510 | atgtggcggccgcTGGAAAAGCGTTTCGGAAGG |
| SR773 | CGCCTTTAAAGCTGAAGTGG |
| SR789 | CAGCTGCGTCTCCGTTTGC |
| SR793 | CCTTCACCATTTAAGACGGAGCAGTAAACAGTTGCTG |
| SR797 | CTGCTCCGTCTTAAATGGTG |
| SR818 | CGGCATGCCGCTACCCGC |
| SR819 | GGGTAGCGGCATGCCGCC |
| SR828 | tttgctcacatgtggcggccgcCAGCCCTGTACACCAGCTC |
| SR829 | gcgcaagaaagaagcttgatATCTCGGCGCCAGGTTTAC |
| SR834 | ccatatcgaaggtcgtcatatgATGGCAATCGCAGACGCTG |
| SR835 | gctttgttagcagccggatctcaGCCGAGGACGGTCATCAG |
| SR836 | GACGTCATCCACTGCCTGTG |
| SR837 | CGACGCATCCTCAACACACC |
| SR851 | TGTGCGCCTTCAATTTGAGC |
| SR852 | GCTCAAATTGAAGGCGCACA |
| SR853 | TAGCCCTTCGTTACCATCGTC |
| HT7 | GCAAACGGAGACGCAGCTG |
| KP235 | CTCCATCACAATTGCCTGCA |
| KP337 | GTGTGGTCGGGCCGTCTAGAA |
| KP342 | TGGTCCGCCGGAACAGATCTTCC |
| KP344 | CTTGTCGTCATCGTCCTTGTAGTCGATGT<br>CGTGATCCTTATAGTCACCGTCATGGTCC<br>TTGTAGTCCATGCCGCTACCCGCCCCA |
| KP345 | GGACGATGACGACAAGGGCAGCAGCCC<br>GCCCAGCCCTTGTAGCAGCAG |
| KP346 | GTCAGCCCTGTTCTGCCC |
| KP347 | AACCCTAAACCCGCTGG |
| qRT_*SULTR2*_Fw | ACGTGGCATGCAGCTCAT |
| qRT_*SULTR2*_Rv | CTTGCCACTTTGCCAGGT |
| qRT_*LHCBM9*_Fw | TGGTGGTGCTTTCCCTTCAGAC |

*Table 3 continued on next page*

*Table 3 continued*

| Primer name | 5'–>3' sequence |
| --- | --- |
| qRT_*LHCBM9*_Rv | TGGACACAACTGCAGGCTTTGC |
| qRT_*HSP22F*_Fw | TGCGCACGCGACATTATCAAAG |
| qRT_*HSP22F*_Rv | GTACAAACCAGCATGCGCTCAG |
| qRT_*VIPP2*_Fw | CATCATGCATTTGGCAGGCTCTC |
| qRT_*VIPP2*_Rv | AATGAGAGGTGCGACGACCAAC |
| qRT_*SNOAL*_Fw | TGCTGTGGGCAACTGTGCATAC |
| qRT_*SNOAL*_Rv | TCCGTGCTTGACGCTACCATTC |
| qRT_*LHCSR3.1*_Fw | CACAACACCTTGATGCGAGATG |
| qRT_*LHCSR3.1*_Rv | CCGTGTCTTGTCAGTCCCTG |
| qRT_*PSBS1*_Fw | TAAACCGTGTATTGGAACTCCG |
| qRT_*PSBS1*_Rv | CTCTGCACGCGGCGTGTT |
| qRT_*CPLD29*_Fw | AACCGGGTCTTCTTCGCCTTTG |
| qRT_*CPLD29*_Rv | GTGTGCCGCCATTCCAAAGAAC |
| qRT_*GBLP*_Fw | CAAGTACACCATTGGCGAGC |
| qRT_*GBLP*_Rv | CTTGCAGTTGGTCAGGTTCC |

Autofluorescence: Excitation 633 nm, Emission filter 670/30 nm, PMT 300, Sensitivity Normal, Pixel size 500 µm. YFP Fluorescence: Excitation 532 nm, Emission filter 555/20 nm, PMT 800, Sensitivity Normal, Pixel size 500 µm. The focal plane parameter was set at 'plus 3 mm' to focus the optics 3 mm higher than the glass plate. This last detail and the thickness of the agar plate (97 ml of liquid agar) were critical parameters to successfully detect the YFP signal.

## Semi-automated identification of YFP mutants through Image-J macroscripts

We used macroscripts in ImageJ64 software (*Schneider et al., 2012*) to quantify the intensity values of colonies in the mutant library (for details, please refer to *Source code 1*). Each plate was imaged in the YFP and in the chlorophyll channel. To orient each image, the bottom two rows of each plate were spotted with characterized positive and negative YFP strains (CrPW1, DCH16, CrPW2, and CrPW3) in the specific order outlined in the scheme below, where (-) denotes lack of YFP signal and (+) denotes presence of YFP signal. The ordering of these colonies conferred a reproducible fluorescent pattern in the YFP channel, which was used to identify the bottom two rows of each image.

**Plate (-Vit)**

| position | 1 | 2 | 3 | 4 | 5 | 6 | 7 | 8 | 9 | 10 | 11 | 12 | 13 | 14 | 15 | 16 | 17 | 18 | 19 | 20 | 21 | 22 | 23 | 24 |
| --- | --- | --- | --- | --- | --- | --- | --- | --- | --- | --- | --- | --- | --- | --- | --- | --- | --- | --- | --- | --- | --- | --- | --- | --- |
| Strain | DCH16 | CrPW1 | DCH16 | CrPW2 | CrPW3 | CrPW1 | CrPW3 | CrPW1 | CrPW2 | DCH16 | CrPW2 | CrPW3 |
| YFP signal | - | - | - | - | - | - | + | + | + | + | - | - | + | + | - | - | + | + | - | - | + | + | + | + |

**Plate (+Vit)**

| position | 1 | 2 | 3 | 4 | 5 | 6 | 7 | 8 | 9 | 10 | 11 | 12 | 13 | 14 | 15 | 16 | 17 | 18 | 19 | 20 | 21 | 22 | 23 | 24 |
| --- | --- | --- | --- | --- | --- | --- | --- | --- | --- | --- | --- | --- | --- | --- | --- | --- | --- | --- | --- | --- | --- | --- | --- | --- |
| Strain | DCH16 | CrPW1 | DCH16 | CrPW2 | CrPW3 | CrPW1 | CrPW3 | CrPW1 | CrPW2 | DCH16 | CrPW2 | CrPW3 |
| YFP signal | - | - | + | + | - | - | + | + | + | + | + | + | + | + | + | + | + | + | - | - | + | + | + | + |

To quantify the YFP intensity of each mutant, a $16 \times 24$ array containing 384 Regions-Of -Interest (ROIs) was constructed for each image in the chlorophyll channel, where all living colonies exhibited signal. The same grid was applied to the corresponding image in the YFP channel. From each ROI on autothresholded images, the maximum intensity in the YFP channel was measured.

To account for variability in the magnitude of the YFP response on different plates (due to slight variations in agar thickness), it was necessary to normalize the YFP intensity of each mutant colony to

the YFP intensity of its parental strain (CrPW1) from the same plate (average of n = 12). Colonies exhibiting YFP fluorescence higher than three standard deviations from the average of all colonies were labeled as potential activators, while colonies with YFP intensities below three standard deviations from the average were labeled as potential silencers.

Of the potential cpUPR silencers, we observed that many of these mutants grew to a larger colony size than the parental CrPW1 strain after 6 days. Their robust growth suggested suppression of vitamin-induced ClpP1 inactivation. To exclude these suppressor mutants, we analyzed the area of all the colonies at 2 and 6 days by measuring the particle area of autothresholded images in the chlorophyll channel. The average colony size increase was 1.38-fold for the 10000-plus colonies analyzed. Candidate colonies that increased 2-fold in colony area (more than one standard deviation away from the average) were regarded as suppressors. Of the remaining silencing candidates, *mars1-1* exhibited the most attenuated YFP response in the presence of Vit.

We indicate *mars1-2* with an asterisk (*) in *Figure 2A* because this mutant was identified in a secondary screen but its position in the original mutant library was lost due to contamination of the plate. To evaluate the YFP response and colony size of *mars1-2* in the context of the entire mutant library, we re-spotted *mars1-2* on a fresh agar plate containing the characterized cpUPR reporter strain and the other positive and control strains in the bottom two rows as described above. The normalized YFP intensity of *mars1-2* in the absence and presence of Vit was then mapped onto the quantification of the original mutant colonies.

## Genomic DNA extraction

With the single exception of the DNA samples submitted for whole genome sequencing, all the other genomic DNA (gDNA) extractions were performed as described below. A 6 ml aliquot of a liquid TAP culture in mid-log phase were spun down, and the media was decanted. The pellet was resuspended in 400 µl of water and then 1 vol of 2x DNA lysis buffer was added (200 mM Tris HCl pH 8.0, 6% Sodium Dodecyl Sulfate (SDS), 2 mM EthyleneDiamineTetraAcetic acid (EDTA). To digest proteins, 5 µl of 20 mg/ml proteinase K (Life Technologies) was added and allowed to incubate at Room Temperature (RT) for 15 min. 200 µl of 5M NaCl was then added and mixed gently. Next, to selectively precipitate nucleic acids, 160 µl of 10% cetyltrimethyl ammonium bromide (CTAB) (EMD Millipore) in 0.7 M NaCl was added and allowed to sit for 10 min at 65℃ with gentle agitation. Two or more consecutive rounds of DNA extraction using ultrapure phenol:chloroform:isoamyl alcohol (25:24:1, v/v/v) were performed to achieve a clean interphase. Then, the upper aqueous phase was retained and mixed with 1 vol of 2-propanol (Sigma). This was mixed gently for 15 min at RT. Then it was spun down for half an hour at 21,000 x g at 4℃. The supernatant was removed and 1 vol of ice-cold 70% ethanol was added and mixed with the pellet. This mixture was spun down for 15 min at 21,000 x g. The supernatant was removed and the DNA precipitate was dried in a speed-vac for about 10–25 min and resuspended in 40 µl of nuclease-free water (Ambion). To ensure complete removal of any potential RNA contamination, in most cases, the gDNA prep was then subjected to in-solution ribonuclease treatment using Rnase A/Rnase T1 mix (Thermo Fisher Scientific) according to manufacturer's instructions. Finally, the gDNA was quickly repurified through an additional round of DNA extraction using ultrapure phenol:chloroform:isoamyl alcohol (25:24:1, v/v/v) and 2-propanol precipitation as described above.

The purity of the gDNA prep was assessed by Nanodrop (Thermo Fisher Scientific), ensuring absorbance ratios at 260/280 nm and 260/230 nm to be ~1.8 and ~2.0, respectively, prior to using the gDNA preparation for most of the follow-up applications. For the pooled (whole genome sequencing) DNA samples, the genomic DNA extraction was performed with the following protocol adapted from the Qiagen, DNeasy Plant Mini Kit using its proprietary buffers (Buffer P3, AW1, AW2, AE). A 25 ml culture of each progeny was grown for ~2 days to about $3 \times 10^6$ cells ml$^{-1}$. Cells were then pelleted and resuspended in 0.5 ml of SDS-EB lysis buffer buffer (50 mM Tris-HCl, 200 mM NaCl, 20 mM EDTA, nuclease-free H$_2$O, 2% SDS, 1% Polyvinylpyrrolidone -average molecular weight = 40,000-, 1 mg/ml of proteinase K) and allowed to incubate for ~10 min. One volume of phenol:chloroform:isoamyl alcohol (25:24:1, v/v) was added and mixed vigorously. The mixture was spun at 13,500 x g for 5 min at RT. The upper phase was transferred into new Eppendorf tubes and 5 µl of 100 mg/ml of RNase A, was added and incubated at RT for 30 min. The lysate was mixed with 130 µl aliquot of Buffer P3 and incubated on ice for 5 min. This mixture was spun at 18,400 x g for 5 min at RT. The lysate was transferred to the QIAshredder Mini column and spun for 18,500 x g

for 2 min. The flow-through fraction was transferred into a new tube and 1.5 volumes of Buffer AW1 was added to the cleared lysate and mixed well. This mixture was then transferred to a DNeasy Mini column and spun at 7600 x g for 1 min. The flow-through fraction was discarded and this step was repeated for any remaining mixture. The DNeasy Mini column was transferred to a new collection tube and 500 µl of Buffer AW2 was added, centrifuged at 7600 x g for 1 min and the flow-through discarded. Another 500 µl of Buffer AW2 was added and centrifuged at 18,400 x g to dry the membrane. The column was then transferred to a new Eppendorf tube. 90 µl of Buffer AE was added onto the DNeasy membrane and incubated for 5 min at RT. The DNA was eluted by centrifugation at 7600 x *g* for 1 min. This step was repeated again, using the 90 µl of Buffer AE collected after the first centrifugation. The quality of the DNA samples was assessed by Nanodrop as described above. The DNA samples were then stored in −20°C until use.

## Single-colony LEAP-seq to identify insertion sites in *MARS1* mutants

The protocol was optimized for single-colony DNA sequencing from the original protocol (*Li et al., 2016*). A pure genomic DNA preparation was assured by running the DNA on a 1.5% agar gel prior to starting. A single-stranded DNA fragment was generated by extending a biotinylated primer from the cassette to the flanking DNA using either primer oMJ598 or primer oMJ1234, which anneals to the 3' or 5' end of the mutagenic cassette, respectively. The linear extension mix was set in the following way: 500 ng of gDNA, 2 µl of 0.25 µM of biotinylated primer, 0.5 µl of Phusion Hot Start Polymerase (Thermo Fisher Scientific), 10 µl of Phusion GC buffer, 3 µl of DMSO, 1 µl of 50 mM MgCl$_2$, 1 µl of 10 mM dNTPs. Prior to starting the mix, the GC buffer was thawed, heated to 95°C for five minutes, vortexed and then put back on ice until the solution became completely clear. The linear extension reaction was carried out in a thermocycler with the following program: Stage 1) 98°C for 3 min; Stage 2) 98°C for 10 s, 65°C for 30 s, 72°C for 18 s (40 cycles). This program was run twice and, in between the first run and the second run, 0.5 µl of Phusion Hot Start Polymerase was added. The Dynabeads kilobase Binder Kit (Thermo Fisher Scientific) was used to purify the linear extension product. For each reaction, 8 µl of streptavidin-coupled magnetic beads) were transferred into an Eppendorf tube and washed in 100 µl of phosphate-buffered saline (PBS) up to four times using a DynaMag magnet (Thermo Fisher Scientific). The Dynabeads were then washed once more in 20 µl of binding solution and gently resuspended in 100 µl of binding solution, pipetting up and down only a few times. Next, the beads were transferred in the PCR tube from the linear extension reaction described above. To allow efficient binding of the linear extension product to the streptavidin-couple magnetic beads, the samples were incubated overnight at RT on an overhead-rotating platform. The following day, the linear extension product was isolated and ligated to a single-strand DNA adaptor, per the following procedure. The beads were washed three times with 100 µl of PBS allowing 8 min incubation in between washes. At the end of the final wash, we ensured that all PBS was carefully removed to avoid interference with the ssDNA ligation reaction. A 20 µl ssDNA ligation reaction was added and gently mixed with the magnetic beads. The ligation mix contained 11.25 µl of H$_2$O, 1 µl of 25 µM ssDNA adapter primer (oMJ619), 1 µl of 50 mM MnCl$_2$, 4 µl of 5 M betaine, 2 µl of CircLigase II reaction buffer, 0.75 µl of CircLigase II (Epicentre). The beads were transferred to the thermocycler, which was pre-heated to 60°C for 10 min. This mixture was incubated for 1 hr at 60°C. The beads were then washed three times with 100 µl of PBS as described above. Next, the ssDNA was converted to a dsDNA using primers annealing to the ligated adaptors at the ends of the ssDNA sequence. 1 µl of 25 µM of Primer 1 (see below), 1 µl of 25 µM of Primer 2 (oMJ621), 0.5 µl of Phusion HotStart, 10 µl of Phusion GC buffer, 32.5 µl of H$_2$O, 3 µl of DMSO, 1 µl of 50 mM MgCl$_2$, 1 µl of 10 mM dNTPs and the template DNA (beads) were mixed together. Primer one depended on whether the original extension from the cassette was in the 5' or 3' orientation. If the 3' cassette flanking primer (oMJ598) was used during the linear extension, primer T3_3'oMJ016c 11/23 was chosen. Instead, when the 5' cassette flanking primer (oMJ1234) was used during linear extension, oMJ1239 was chosen. Primer 2 (oMJ621) annealed to the ligated adaptor. Both primers were designed to contain a binding site for the mutagenic cassette and a binding site for a T3 sequencing primer. The following amplification program was used: Stage 1) 98°C for 3 min, Stage 2) 98°C for 10 s, 63°C for 25 s, 72°C for 20 s (10 cycles), Stage 3) 98°C for 10 s, 72°C for 45 s (13 cycles). The dsDNA products were then run on a 1% gel. The DNA smears were cut out of the agarose gel, purified using NucleoSpin Gel and PCR Clean-Up Kit (Takara) according to manufacturer's instructions and subjected to Sanger Sequencing using a standard T3 sequencing primer. Finally, to identify

the insertion site of the mutagenic cassette, the sequencing results were blasted in Phytozome, v5.5. The sequence of the aforementioned primers can be found in *Table 3*.

## Mating and tetrad analysis

Note: *mars1-1* and *mars1-2* proved difficult to mate due to their genetic background. Therefore, extra measures were taken to increase the efficiency of mating. *mars1-3* and *mars1-4*, obtained from the Jonikas library, did not have this problem, therefore, these extra measures were not taken. The following protocol will indicate the differences.

Cells were re-streaked onto fresh TAP agar and incubated in low light (<15 µmol photons m$^{-2}$ s$^{-1}$) for five days. They were then transferred onto TAP agar containing 1/10 of the usual NH$_4$Cl concentration and kept in this medium for four-five days under moderate light (~40 µmol photons m$^{-2}$ s$^{-1}$) to induce starvation. The gametes from each strain were then resuspended in a 24-well sterile transparent plate (Costar) using 150–200 µl of water or M-N/5 solution till a dark green resuspension is obtained. M-N/5 solution was used for the *mars1-1* and *mars1-2* backcrosses, and water was used for *mars1-3* and *mars1-4* backcrosses. M-N5 solution contained 1 ml of 10% sodium citrate, 0.2 ml of 1% FeCl$_3$, 0. 2 ml of 4% CaCl$_2$, 0.34 ml of 10% K$_2$HPO$_4$, 0.2 ml of 10% KH$_2$PO$_4$, 0.2 ml of Hutner's Trace elements (Chlamydomonas Resource Center), H$_2$O to 1.25 liter. The solution was autoclaved in 100 ml aliquots per bottle. The plate was then transferred to a shaker under moderate light (~40 µmol photons m$^{-2}$ s$^{-1}$) and allowed to mix for ~1 hr. Gametes of the opposite mating type were mixed (~100 µl per gamete) in a separate well and the plate was placed under light with no shaking. For the *mars1-1* and *mars1-2* strains, after one hour of mixing, dibutyryl cyclic AMP (Sigma) was added to each mating mix to a final concentration of 30 mM. Mating efficiency was checked periodically (every 15–30 min) for fusion events and quadriflagellate formation. The gametes were mated for ~3 hr. Aliquots (100 µl) of the mating mixture were plated into TAP 4% agar. Plates were exposed to light (~50 µmol photons m$^{-2}$ s$^{-1}$) overnight and the next day wrapped in aluminum foil. After ~1–2 weeks, the vegetative cells were scraped off using a small rectangular soft razor blade (Personna, .009'', two-facet aluminum blade) with gentle pressure on the agar. Zygotes adhere to the agar surface and can be recognized under a light microscope due to their darker and larger appearance. A 100 µl aliquot of liquid TAP medium was then added on top of the zygotes and a more rigid scalpel (Feather, N.2) was used to scrape the zygotes off the agar. A line was drawn onto the center of a fresh TAP 1.5% agar plate, and the zygotes were spotted along this line. The cells were then allowed to dry. For *mars1-1* and *mars1-2,* but not for *mars1-3* and *mars1-4,* vegetative cells were killed by treating the plate with chloroform vapor for ~15–30 s. The plate was incubated under light (~80 µmol photons m$^{-2}$ s$^{-1}$) overnight to 1.5 days. At 24°C, germination typically occurred after ~20 hr. Under the dissection scope, tetrads and octads were found and dissected. Incomplete tetrads, full tetrads and octads were then re-arrayed onto TAP agar in a 96-array format and then replicated onto the appropriate drug resistances. When necessary, mating type-specific PCRs (*Werner and Mergenhagen, 1998*) were carried out to ensure that the progeny were in fact due to a sporulation event and were not mistakenly parental strains.

## Check-PCRs on genomic DNA to verify the causative mutation in *mars1-1*

Genomic DNA from progeny derived upon crossing *mars1-1* to CC-124 was obtained as outlined in 'Genomic DNA extraction' section. The insertion of the mutagenic cassette in the *MARS1* locus was verified by PCR by using primers SR773 and T3_5'_oMJ016c 11/24, which anneal to exon 17 of *MARS1*, and to the 5' side of the mutagenic cassette, respectively. The PCR reaction was run on 1–1.5% agarose, cut out of the agarose gel, purified using NucleoSpin Gel and PCR Clean-Up Kit (Takara) according to manufacturer's instructions and subjected to Sanger sequencing to verify the expected sequence identity. The sequence of the aforementioned primers can be found in *Table 3*.

## Whole Genome Sequencing (WGS)

Progeny derived upon crossing *mars1-2* x CC-124 were re-arrayed onto 96-well plates and replicated onto TAP supplemented with hygromycin, paromomycin or spectinomycin to determine the segregation patterns of the progeny. Mating type-specific PCRs were performed on almost all progeny (*Werner and Mergenhagen, 1998*). The progeny was then tested in high light and by Vipp2

immunoblot analysis two or three times to determine which one had silencing vs. wild-type phenotypes. Genomic DNA was extracted as described above in the section 'Genomic DNA extraction' and was subsequently pooled per the progeny's phenotype, i.e. *mars1*-like or WT-like. Additional pools containing the parental strains were also analyzed likewise. The size of the pools of gDNA were of different proportions depending on the amount of progeny in that group (WT vs. *mars1*-like). The pooled gDNA was then fragmented using Covaris and Bioruptor Pico. The sequencing libraries were prepared with the aid of the PrepX DNA library kit (Takara). One cycle of PCR was used to linearize the library molecules. Fragment analyzer traces and Qubit values were assessed for each sequencing library as quality control checks. Sequencing was performed on the HiSeq2500 Rapid sequencer. The *C. reinhardtii* reference genome was downloaded from Phytozome, v5.5 onto the Geneious software (*Kearse et al., 2012*). The reads from each library were then aligned to the reference genome.

## Check-PCRs on genomic DNA to verify the causative mutation in *mars1-2*

Genomic DNA from progeny of meiotic tetrads derived upon crossing *mars1*-2 to CC-124 was obtained as outlined in 'Genomic DNA extraction' section. The *MARS1* locus was amplified by using primers SR789 and KP235, which anneal to exon 15 and the intron 19 - exon 20 junction of *MARS1*, respectively. The *MARS1* deletion locus was amplified by using primers KP346, which anneals to intron 1 of the *MARS1* gene, and KP347, which was derived from the a WGS read found only in the '*mars1*-like' progeny pool. The KP347 primer sequence is a hybrid of telomeric sequence and *MARS1* gene sequence - this sequence seems to have arisen after a genomic deletion at the end of chromosome 16 in *mars1-2*. The PCR reactions were then purified and sequenced as described above. The sequence of the aforementioned primers can be found in *Table 3*.

## *MARS1* gene cloning

A *MARS1* 'midigene' was generated by amplifying four different portions of this gene either from gDNA or cDNA using KOD Hot Start DNA Polymerase (Thermo Fisher Scientific) or Phusion Hotstart II polymerase (Thermo Fisher Scientific). In particular, the region spanning the promoter, the 5'UTR and the first 5 exons of this gene was amplified from gDNA using Phusion polymerase and the following primers: SR828 and SR818; the region spanning exon 5 to exon 15 was amplified from gDNA using KOD polymerase and the following primers: SR819 and HT7; the region spanning exon 15 to exon 28 was amplified from cDNA using KOD polymerase and the following primers: SR789 and SR797 and the 3'UTR was amplified from gDNA using KOD polymerase and the following primers: SR793 and SR829.

All PCR products were gel extracted and purified as described above in the section regarding the pPW3217 cloning. Next, these 4 PCR fragments were mixed with a purified and linearized and pRAM118/pPW3216 vector, previously digested by EcoRV and Not1, and incubated in presence of the In-Fusion reagents (Takara) as per manufacturer's instructions. The resulting plasmid is notated as pPW3218. The sequence of the aforementioned primers can be found in *Table 3*.

The Phytozome v5.5 *MARS1* transcript annotation is Cre16.g692228.t1.1.

## *MARS1* gene tagging

### Generation of *MARS1-A* transgene (pPW3219)

To insert a 3x-Flag epitope after Arg1167 of the Mars1 protein sequence, a dsDNA gene block was synthetized by IDT with the following sequence:

GGTACGACGGCTGGGCTGGGGCGCCGGCGTCCGCCCCCTGCTCCCAAGTTGTCATTGCCA
TCAGCGGCAGGCGTGGGGCATCGGTTGCAGCCGGTTTCGCCGGCTTCCACCGTGTCCGGGC
TTCCTTGGGGCCAGGCTGCGCACCCGTCGCACACAGCTCCGCAGCTGCTCCGCGCGCCGCA
TCAACAGTTCGGGAGCATTGCTGCAGCAGCAGCGAGTGCCGCTCCAAGCTGGGCTGCAAGC
TCGAGCAGTGCTGCACTGGTAGCAGCTGAGCTTGGGGCAGCTGCAGTAGCAGCTGCAGCGG-
CAGCGGCGGCTTGCACCTCGGCTTCTGTGGGCGTAGACCCGGGATCATCGGCTGTGAACACACA
TGCCGCCGCTGCCGTCGCTCCCTCCATGTGGAAGGCGGCCCTGCTGGCACCCGGTGGGGAGGC
TCCGCGTGGGAATGGCTCAGCCCGGAGCAGCTTTGAGGCGGGGGGAGCCATCACCG
TCGGAGCGGGCACGCAGGCAGCAAGAGCAGCTGGCAGCGGCGGCAGCA

TCGGAGGGGCGGCCTGCGGCTAGCACAGGCCAGAAGCCGGCAGCGTCTTCGGCTGTTGCAAC-
CACGTCCAGCTCAACCTCCACTGCCAGGCGGAGAGACCAGCAGGGTAACTCGCAGTCACGGC-
CAGTAGTGGAGCGTGGTTCGGGCGGTggtggctccgactacaaggaccatgacggtgactataaggatcacgacatc-
gactacaaggacgatgacgacaagggtggcggcggcagtGGGAGAGGTGCTGCTCGCGGCGGCATGTCCA-
CACGGGGCGGGGGAACTGGAGGCCGGGGCAGTGGACGCCTGTTCGGCAGAGGACGTGGGA-
GACTGGACCGCGGAGATGACGACAACGGTTACGCGGAGGAGAACCAGCCATCTGCAA
TCGGCGCCGCGAGCAATTCCGAACAGCTGGAGCACGGCCGACAGCGCCGTGAGGG
TGCGGGAGGTGACGGCGCTCACGAGCAGGGGGCTGGGGCTGCCAGCAGCTCGGCCCAGCC-
CAAGCTGCCTCTCGCAACTACGGGCACAGCAGCTGCCTCGGAGCACTCTGGCGCTGTTGATTC
TTCAACAGCTACCGCCGGCGCTCCCGACGCAGCTAGCCCT

Lower case letters indicate the Flag epitope-encoding insertion. Note that, in this fragment of DNA, often a *MARS1* codon ending with G or C was mutated to a synonymous codon ending in A or T as a strategy to decrease the amount of GC in the sequence.

Next, this gene block was mixed with a purified and linearized pPW3218 vector, previously digested by MreI and dephosphorylated by Calf Intestinal Phosphatase (CIP, NEB), and incubated in presence of In-Fusion reagents (Takara) as per manufacturer's instructions. The resulting plasmid is notated as pPW3219.

## Generation of *MARS1-D* transgene (pPW3222)

To insert a 3x-Flag DNA sequence downstream of the second in-frame translation start codon, ATG (ii), (actual position: Met139), two partially overlapping regions of the *MARS1* gene were amplified by PCR using as template pPW3218. The following primer pairs were used: KP337/KP344 and KP345/KP342, to enable the insertion the 3x-Flag epitope after Met139 of the Mars1 protein. Next, these two PCR products were gel-purified as already described above and mixed with a purified and linearized pPW3218 vector, previously digested by AvrII and XbaI. These three DNA fragments were incubated in presence of In-Fusion reagents (Takara) according to manufacturer's instructions. The resulting plasmid is notated as pPW3222. The sequence of the aforementioned primers can be found in *Table 3*.

## Generation of *MARS1-E* transgene (pPW3223)

The insertion of a 6x-Flag epitope after Leu402 of the Mars1 protein sequence was a fortuitous byproduct of a cloning strategy in which a 3x-Flag dsDNA sequence inserted twice rather than once. This 3x-Flag dsDNA, with BglII-compatible sticky ends, was generated upon annealing of the two following single-stranded DNA fragments:

3x-Flag_UP:
GATCTGGACTACAAGGACCATGACGGTGACTATAAGGATCACGACATCGACTACAAGGACGA
TGACGACAAG;

3x-Flag_DOWN:
GATCCTTGTCGTCATCGTCCTTGTAGTCGATGTCGTGATCCTTATAGTCACCGTCATGGTCCTTG
TAGTCCA

and it was mixed with a linearized and purified pPW3218, digested by BglI and dephosphorylated by Calf Intestinal Phosphatase (CIP, NEB) and incubated in presence of In-Fusion reagents (Takara) according to manufacturer's instructions. A clone containing a plasmid with a double insertion of the 3x-Flag DNA sequence at the BglII site was identified by analytical digestion and confirmed by Sanger sequencing. The resulting plasmid is notated as pPW3223

## Generation of *MARS1-A KD*, *MARS1-D KD* and *MARS1-E KD* transgenes (pPW3224, pPW3226 and pPW3225)

pPW3226, pPW3224 and pPW3225 were all constructed using a three-piece In-Fusion assembly technique. For the kinase-dead version of the *MARS1-E* transgene, the backbone pPW3223 was digested with EcoRV and BstXI and the D→A point mutation was introduced by generating two DNA fragments by PCR using pPW3223 as template and Phusion Hotstart II polymerase (Thermo Fisher Scientific) with the following primers pairs: SR852/SR853 and SR829/SR851. The three DNA fragments were incubated in presence of In-Fusion reagents (Takara) according to manufacturer's

instructions. A clone containing the D→A mutation was identified by Sanger sequencing. The resulting plasmid is notated as pPW3226.

Similarly, for the kinase-dead (KD) version of the *MARS1-A* transgenes, the backbone pPW3219 was digested with EcoRV and BstXI. The primers to introduce the D→A mutation and the strategy to generate and identify the correct plasmid clone were the same as for pPW3226. The resulting plasmid was notated as pPW3224.

To introduce the D→A point mutation in the *MARS1-D* transgene, two partially overlapping regions of the *MARS1* gene were amplified by PCR using as template pPW3218 and the following primer pairs: KP337/KP344 and KP345/KP342 as already described above in the case of pPW3222 cloning. Next, these two PCR products were mixed with a purified and linearized pPW3226 vector, previously digested by AvrII and XbaI. These 3 DNA fragments were incubated in presence of In-Fusion reagents (Takara) according to manufacturer's instructions. The resulting plasmid is notated as pPW3225. The sequence of the aforementioned primers can be found in *Table 3*.

## *MARS1* transgene nuclear integration

Given the large size of the *MARS1* transgene (>10 kbp), a different electroporation protocol was used to successfully integrate each aforementioned *MARS1* transgene into the nuclear genome. The electroporation was performed using a NEPA21 electroporator (Nepagene) (*Yamano et al., 2013*). A 4–8 µl aliquot of purified, non-linearized, plasmid DNA at a concentration of 0.5–1 mg ml$^{-1}$ was used per transformation. In each case, the plasmid DNA was mixed together with 5 µl of a ready-to-use, sheared solution of salmon sperm DNA at a concentration of 10 mg ml$^{-1}$ (Thermo Fisher Scientific) prior electroporation. The electroporation parameters were set as follows: Poring Pulse (300 V; length = 6 ms; Interval = 50 ms; No = 1; D.Rate = 40%; + Polarity), Transfer Pulse (20 V; length = 50 ms; Interval = 50 ms; No = 5; D.Rate = 40%; +/- Polarity). Usually, during the electroporation, the impedance was measured to be around 400–700 ohms. Transformants were isolated on TAP agar containing 20 µg ml$^{-1}$ hygromycin and screened by Flag immunoblot analysis to identify *MARS1* transgene expressors.

## Dual immunofluorescence (IF)

Due to the low abundance of the Mars1 protein, the Alexa Fluor 488 Tyramide SuperBoost Kit (Thermo Fisher Scientific) was utilized to amplify the signal. WT (mock control) and Flag-tagged strains were grown in TAP medium to logarithmic phase and a 5 ml aliquot of each cell cultures was harvested at 500 x g for 5 min at RT. The supernatant medium was decanted and the cell pellet was resuspended in 1.5 ml of PBS. A 35 µl cell suspension was added to wells on a slide pre-treated with 0.1% poly-L-lysine and allowed to adhere for 7 min. To solubilize the chlorophyll and maintain cell structure, slides were incubated in 100% methanol 2 times for 4 min at −20˚C. Excess methanol was removed and the slides were dried for 2 min at RT. Slides were incubated in PBS supplemented with 0.1% Tween 20 (PBS-T) for 10 min to permeabilize the cell before adding the manufacturer-provided 3% H$_2$O$_2$ solution to quench endogenous peroxidase activity for 1 hr. Following three washes with PBS, non-specific signal was blocked with 10% goat serum in PBS (Jackson ImmunoResearch Laboratories) for 1 hr. Samples were incubated with commercial monoclonal mouse anti-Flag antibody (M2 Sigma, F1804, diluted 1:500) in combination with one of three subcellular markers: anti-Histone H3 (Agrisera, AS10 710, diluted 1:500), anti-AtpD (Agrisera, AS10 1590, diluted 1:500), or anti-Nab1 (Agrisera, AS08 333, diluted 1:500), and left overnight at 4˚C in a humid chamber. The remaining steps were performed as per manufacturer's instructions. In brief, samples were washed for 10 min in PBS for a total of three washes. Alexa Fluor 488 poly- horseradish peroxidase (HRP)-conjugated goat anti-mouse and Alexa Fluor 546 goat anti-rabbit secondary antibodies were utilized for detection of the Flag and subcellular marker signals respectively. Alexa Fluor 546 was diluted 1:500 in the manufacturer-prepared Alexa Fluor 488 solution and added to each well for 1 hr at RT protected from light. Slides were washed 3 times in PBS before the amplification step. To amplify the Flag signal, 35 µl of the Tyramide Working Solution was added to the cells and the reaction proceeded for 4 min before an equal volume of Stop Solution was added to end the reaction. Slides were rinsed in 1x PBS and covered with a thin layer of 0.5% low-melting point agarose dissolved in TAP medium before observation by 3D-Structured Illumination Microscopy (3D-SIM).

## 3D-Structural illuminated microscopy (3D-SIM)

The microscopy samples were observed using an Elyra PS.1 SIM microscope (Zeiss) with objective lens alpha Plan-Apochromat 100x/1.46 oil (Immersol 518F/30°C, Zeiss), as described previously (*Iwai et al., 2018*). The fluorophores Alexa Fluor 488 and Alexa Fluor 546 were excited with a 488 nm laser and 561 nm laser, and the fluorescence was acquired through a 495–550 nm and 570–620 nm bandpass filters, respectively. Image acquisition was performed with ZEN software (Zeiss). Each focal plane for 3D-SIM image was captured sequentially by the excitation with the patterned light of 3 rotated angled, each of which contains five shifted phases. The optimal z-interval distance was set to 101 nm. Raw SIM images were processed to reconstruct 3D-SIM images using ZEN software. Extraction of the intensity data was done using the SIMcheck plugin for ImageJ software (*Ball et al., 2015*).

## Cytosolic fractionation

For localization studies, cytosol-enriched and cytosol-depleted fractions were isolated from *mars1-3* cells (a cell-wall deficient strain) complemented with the *MARS1-A* transgene, according to the protocol described below, incorporating guidelines previously described in *Klein et al., 1983* and *Zerges and Rochaix (1998)*. A one liter liquid culture, synchronized by growth in dark-light cycles in minimum media and in early exponential phase ($1–2 \times 10^6$ ml$^{-1}$), was harvested at 3000 x g for 5 min at RT. Upon media removal, the cell pellet was resuspended by gentle hand-shaking of the centrifugation tube without using a pipette in 15 ml of autolysin freshly supplied with 1 mM potassium phosphate buffer (pH 6.0) and 0.5 mg/ml BSA (Solution A). Solution A was pre-warmed at 30°C for 30 min prior to use and, after resuspending the pellet in this solution, cells were transferred to a 200 ml 30°C-prewarmed beaker immersed in a water bath and incubated at 30°C for 50 min. Next, cells were transferred to a 50 ml Falcon tube and collected by centrifugation at 2000 x g for 5 min at RT. The autolysin was quickly removed using a 25 ml plastic pipette and the cell pellet was very gently resuspended in 20 ml of ice-cold Solution B consisting of 5 mM K-phosphate buffer (pH 6.5), 6% PEG (w/w) and 4 mg/ml BSA.

Cells were then transferred to a 32°C-prewarmed beaker immersed in a water bath at 32°C and 80 μl of freshly-prepared 1% digitonin (Calbiochem) (0.004% final concentration) was quickly added and well-mixed with the cell suspension.

Cells were then subjected to two rapid warming-cooling cycles using 32°C pre-warmed or ice pre-chilled beakers to induce plasma membrane rapture and cytosolic protein release without intracellular organelles breakage. Cycling was performed for 2 min at 32°C, 5 min on ice, 1 min at 32°C, 5 min on ice. Next, this suspension of permeabilized cells and released cytosolic proteins was transferred to a 50 ml pre-chilled Falcon tube and centrifuged at 800 x g for 3 min at 4°C. After centrifugation, the cytosol-enriched fraction was further purified from the supernatant fraction, while the cytosol-depleted fraction containing chloroplasts and mitochondria was further purified from the cell pellet.

To remove potential cell debris and obtain a clean cytosolic fraction, the supernatant was subjected to two further consecutive rounds of centrifugation: first, at 5000 x g for 15 min at 4°C and then at 23,000 x g for 1 hr at 4°C. Finally, the cytosol-enriched fraction was precipitated in ice cold acetone containing 10% of trichloroacetic acid (TCA).

To enrich the cytosol-depleted fraction with organelles, the pellet of permeabilized cells was kept in ice and resuspended in 2 ml of ice-cold 2x isotonic solution consisting of 0.6 M sorbitol, 10 mM MgCl$_2$ and 20 mM Tricine pH 7.8. At this point, a rather dark-green aggregate formed in the falcon tube. To resuspend this aggregate, a cut plastic pipette tip was used to gently pipette the pellet up/down for 20 times. Then, 2 ml of ice-cold milliQ water were added to bring the isotonic solution to 1x and the aggregates were further dissolved as described above.

Next, this suspension was loaded on a Percoll step gradient (10 ml 75% Percoll in isotonic solution/10 ml 45% Percoll in isotonic solution/4 ml cell lysate) in a Corex glass tube. The gradient was subjected to centrifugation using the HB4 swinging-rotor at 7000 x g for 15 min at 4°C. Chloroplasts and contaminating mitochondria were recovered from the interface between 45% and 75% Percoll and diluted in 20 ml of 1x ice-cold isotonic solution. The organelles were collected by 5 min centrifugation at 4000 x g at 4°C and, after removing the supernatant, the pellet was resuspended in 2 ml ice-cold isotonic solution and run through a second Percoll gradient as described above. Finally, the cytosolic-depleted fraction was precipitated in ice-cold acetone containing 10% TCA. Denatured

proteins from each fraction were extracted after TCA precipitation, as described below. Prior to gel electrophoresis, the protein content of each fraction was normalized using a bicinchoninic acid (BCA) assay as described in the section below.

## Denaturing protein extraction, immunoblot analysis and BCA assay

Proteins were extracted from whole cell lysate using a denaturing SDS extraction protocol for all experiments except for immunoblots that include the Mars1 protein, in which case TCA precipitation was used. For the SDS protein extraction, cells from a 5 ml culture in exponential growth phase were harvested at 3000 x g for 5 min and resuspended in 150 µl of SDS-lysis buffer (100 mM Tris-HCl pH 8.0, 600 mM NaCl, 4% SDS, 20 mM EDTA, freshly supplied with Roche Protease Inhibitors). Samples were vortexed for 10 min at RT and centrifuged at maximum speed for 15 min at 4°C to remove cell debris. The supernatant, containing a total extract of denatured proteins was transferred to a new Eppendorf tube, a 5 µl aliquot was saved for BCA quantification and 1/4 vol of 5X SDS-loading buffer (250 mM Tris-HCl pH 6.8, 5% SDS, 0.025% bromophenol blue, 25% glycerol), freshly supplied with 500 mM DTT or 5% of 2-mercaptoethanol prior use, was added to the extract and denatured at 37°C for >30 min. For the TCA precipitations, cell pellets were resuspended in 1 ml of 10% TCA in acetone, freshly supplemented with 0.5% β-mercaptoethanol. Samples were vortexed for 10 min at 4°C then left at −20°C for 1–2 hr for efficient protein precipitation. Samples were centrifuged at maximum speed for 10 min at 4°C and the TCA solution was carefully aspirated. Three washes with 1 ml of cold 100% acetone were performed (5 min of vortexing followed by 5 mins of maximum speed centrifugation) and the remaining pellet was dried for 5–10 min before resuspension in Lysis Buffer (same as above), achieved through vigorously shaking of the Eppendorf tube with the aid of a vortex at RT or with the aid of a thermomixer at 50°C for 10–15 min. The resuspended protein extract was isolated by a quick centrifugation and was transferred to a new Eppendorf tube. A 5 µl aliquot was saved for BCA quantification and 1/4 vol of 5x SDS loading buffer was added to the rest and denatured at 37°C for at least 30 min. Immunoblot analysis was performed on 20–60 µg of denatured protein extract. Proteins were separated by SDS-PAGE using Mini-PROTEAN or Criterion Precast Gels) (Bio-Rad) and transferred onto Protran nitrocellulose membrane, 0.2 µm pore (Perkin Elmer). Non-specific signal was blocked with PBS-T supplemented with 5% instant nonfat dry milk (Carnation, Nestlè) for 1 hr at RT or overnight at 4°C. All primary and secondary antibodies were diluted in this blocking buffer. The following antibodies (at the indicated dilution) were used for this publication: monoclonal mouse anti-Flag (1:3,000) (M2, Sigma F1804), monoclonal mouse anti-α-tubulin (1:5,000) (Sigma #T6074), polyclonal rabbit anti-DnaK (provided by Jean David Rochaix) (1:10,000) (H.Naver, K.Wilson and J.D.Rochaix, unpublished results) (*Dauvillee, 2003*), polyclonal rabbit anti-RpoA (1:10,000) (*Ramundo et al., 2013*), polyclonal rabbit anti-ClpP1 (provided by Francis-André Wollman and Olivier Vallon) (1:5,000) (*Majeran et al., 2000*), polyclonal rabbit anti-Hsp22E/F (provided by Michael Schroda) (1:5,000) (*Rütgers et al., 2017*), polyclonal rabbit anti-Vipp2 (1:3,000) (raised against a –CDPLERELEELRRRARE- peptide, developed during this study by Yenzym, South San Francisco), polyclonal rabbit anti-Aox1 (1:2,000) (Agrisera AS06 152), polyclonal rabbit anti-Sultr2 (provided by Arthur Grossman) (1:3,000) (*Pootakham et al., 2010*), polyclonal rabbit holo-Rubisco (provided by Jean David Rochaix) (1:10,000) (*Borkhsenious et al., 1998*), polyclonal rabbit anti-Hsp90C (1:10,000) (Agrisera AS06 174) and anti-Histone H3 (1:10,000) (Agrisera AS10 710). To detect the primary antibodies, HRP-conjugated anti-rabbit and anti-mouse secondary antibodies (Promega) were used at dilution 1:10.000 in PBS-T supplemented with 5% instant nonfat dry milk for 1 hr at RT. In between the incubation with primary and secondary antibody and after the incubation with the secondary antibody, three washes of about 10 min each time, at RT, were performed using PBS-T supplemented with 5% instant nonfat dry milk. The membranes were quickly rinsed three times with milliQ-water and a luminol-based enhanced chemiluminescence (ECL) method was applied to develop the signal. For most immunoblot analysis, the SuperSignal West Dura Extended Duration Substrate kit (Thermo Fisher Scientific) was used according to manufacturer's directions. By contrast, for Flag immunoblot analysis to detect Mars1 protein, the SuperSignal West Femto Maximum Sensitivity Substrate kit (Thermo Fisher Scientific) was chosen, given the low expression level of this protein. The ECL signal was detected with the LI-COR Odyssey imaging system or using clear-blue X-ray films (CL-Xposure, Thermo Fisher Scientific).

To carry out the BCA assay, 5 µl of protein extract was added to 200 µl of BCA/copper sulfate solution (1:50 dilution of 4% $CuSO_4$ into BCA solution, Sigma) and incubated at 50°C for 5 min.

Protein concentration was estimated by measuring the absorbance at 562 nm and comparing it to a BSA standard.

Note: for *Figure 6—figure supplement 1B*, the denaturing protein extraction was carried out as follows: cell cultures started in 7 ml of TAP were grown to mid-log phase and subsequently spun down at 3000 x g for 8 min. The pellets were resuspended in 150 µl of TAP. Then, an equal volume of 0.2M NaOH was added to the pellets, vortexed at RT for 5 min and pelleted at 15,000 x g for 5 min. The supernatant was removed, the pellet was resuspended in 280 µl of SDS samples buffer (0.06 M Tris-HCl pH 6.8, 5% glycerol, 2% SDS, 4% 2-Mercaptoethanol, 0.0025% bromophenol blue), boiled for 5 min and then pelleted again. A 28 µl aliquot was loaded onto a Criterion gel.

## High light (HL) assays

The following protocol was used for the HL assay described in *Figure 4A–B*. Slight modifications were applied during the HL assays described in *Figure 6E* and will be underscored below. Liquid cultures were started from TAP plates that had all been grown in dim light and in the same conditions. A small slab of cells was taken from the agar plate and resuspended in 28 ml of TAP media in 50 ml falcon tubes (Sarstedt). The Olympus Plastics brand product line from Genesee Scientific was avoided because there was a higher propensity for the cells to adhere to this plastic material. An equal slab of cells was used for each culture to approximate the same level of starting cells. Typically, the strains had been growing in fresh TAP agar for ~3–5 days. The cells were pre-conditioned in low light (~20–50 µmol photons $m^{-2}$ $s^{-1}$) for about 38–44 hr. For the HL assays shown in *Figure 6E*, the cells were preconditioned for slightly longer period,~3 days. The chlorophyll concentration of the cell cultures was measured using the methanol extraction method as described in *Porra et al. (1989)*. At the above described time points, it was found to be ~13–18 µg/ml (HL assays described in *Figure 4A–B*), or ~25 µg/ml (in the HL assays described in *Figure 6E*). Cell cultures were then equally diluted to ~10 µg chlorophyll $ml^{-1}$ (in the case of the HL assays described in *Figure 4A–B*), or to 7 µg chlorophyll $ml^{-1}$ (HL assays described in *Figure 6E*). Chlorophyll concentrations were confirmed and, if needed, re-adjusted after dilution before HL was started. The final volume of cell culture used for high light treatment was ~26 ml in 50 ml Falcon tubes. During the high light treatment, the distance between the light source (Phlizon 2017, 2000W Plant LED Growth light) and the shaker was set to 25 cm. The HL intensity was measured at 1100 µmol photons $m^{-2}$ $s^{-1}$ at the beginning of the experiment but was reduced to ~900 µmol photons $m^{-2}$ $s^{-1}$ by the end of the experiment. On the right and left side of the shaker, two fans were turned on to keep the samples at RT and a Smart Sensor (SensorPush) was used to monitor temperature in real-time. Typically a 4˚C increase in temperature (from 24.5˚C to about 28.5˚C) was measured after the cultures were shifted from the dim light growth setup to this HL setup. The light intensity at each position of the culture on the shaker was measured to ensure cells were getting the same number of photons (~50,000 LUX). The cultures in the Falcon tubes were taped (clear tape) onto the shaker and the shaker was set at 150 rpm. Chlorophyll measurements were taken at multiple time points of HL treatment (during HL assays described in *Figure 4A* and *Figure 4—figure supplement 1A*) and after 27 hr or 50 hr of HL treatment (during HL assays described in *Figure 4B* and *Figure 6E*, respectively). Serial dilutions performed on cultures before and after the HL treatment were spotted onto TAP plates. Photographs of these plates were taken over time to track cell recovery.

## Metronidazole assays

For immunoblot analyses, as shown in *Figure 4D*, cell cultures were grown in liquid TAP medium in a 50 ml Falcon tube for about two days to a chlorophyll concentration of 11 µg/ml in a volume of 30 ml. Cells were spun down and the pellet was resuspended in 1.5 ml of TAP. 0.3 ml of the resuspended pellet was then added to 10 ml of TAP with or without 1.1 mM metronidazole (Sigma). These cultures were then placed under white light (20–50 µmol photons $m^{-2}$ $s^{-1}$) for 12–15 hr and then spun down and saved at −20˚C or directly used for denaturing protein extraction as described above.

For growth tests on TAP agar supplemented with metronidazole, as shown in *Figure 4C* and *Figure 4—figure supplement 1B–E*, strains were either manually re-streaked or robotically replicated from TAP agar plate to +/- 1.5 mM metronidazole TAP agar plates.

For dilution spot tests, shown in *Figure 6D*, cells were freshly inoculed from a 3-4 days old agar plate and grown in in falcon tubes for 2 days in a starting volume of 30 ml of TAP. After these 2 days of preconditioning, chlorophyll concentrations were normalized to be at ~18 µg/ml and serial dilutions of 1.5-fold were done in liquid TAP using a 96-well plate. Finally, 6 µl cells from each dilution series were spotted onto TAP agar plates with or without 2.2 mM metronidazole. Photographs were taken periodically to track growth over time.

Metronidazole is rather insoluble in aqueous solutions; therefore, it was always added (as powder) directly to the autoclaved liquid TAP medium at final concentrations of 1.1 mM, 1.5 mM or 2.2 mM.

## Flag affinity purification and mass spectrometry (MS) analysis

Mars1-D and Mars1-D KD strains were subjected to metronidazole treatment for 15 hr. Culture were then harvested and subjected to Flag affinity purification followed by MS analysis according to the protocol described by {Mackinder, 2017 #30} and publically available through this link: https://docs.google.com/viewer?a=v&pid=sites&srcid=ZGVmYXVsdGRvbWFpbnxjaGxhbXlzGF0aWFsaW50ZXJJhY3RvbWV8Z3g6NzlkNjUzMTM0ZWEyYmI5. Preparation of samples for MS analysis and processing of MS raw data was performed by the Stanford Mass Spectrometry Facility in Palo Alto.

## RNA extraction

A 10–ml aliquot of a cell culture in early-mid exponential phase ($1–5 \times 10^6$ ml$^{-1}$), was harvested at 3000x g for 5 min at RT. After decanting the media, 1 ml of Trizol Reagent (ThermoFisher Scientific) was added to the cells. The cells were lysed in sample by vigorous shaking with the aid of a vortex for 5–8 min. Chloroform (1/5 vol,~200 µl) of was then added to the lysate and the tube was vigorously shaken up and down by hand for 60 s. The sample was then centrifuged at 11,000 x g for 7 min at RT. The upper aqueous phase (~350 µl) was removed with care to not draw any of the organic layer, transferred in a nuclease-free 1.5 ml tube (Ambion) and mixed well with 1 vol of 100% ethanol (Fisher Scientific) at RT. From this point onwards, the RNA purification was carried out with the aid of the Direct-zol RNA MiniPrep Plus kit (ZymoResearch) following the manufacturer's protocol, including in-column DNase I treatment prior RNA washing and elution steps.

## Analysis of gene expression by RT-PCR

A semi-quantitative RT-PCR was carried out to qualitatively determine the presence or absence of *MARS1* gene transcripts. Total RNAs were extracted as described in the previous section. Complementary DNA was synthesized from 1 µg of total RNA using PrimeScript 1 st strand cDNA Synthesis Kit (Takara) as per manufacturer's instructions. Subsequently, RNA/DNA hybrids were removed by ribonuclease H treatment (NEB) as per manufacturer's instructions. cDNAs were diluted two-fold and 1–2 µl were used as template for a 20 µl PCR reaction by Phusion High-Fidelity DNA Polymerase (Thermo Fisher Scientific) with the following parameters: initial melting 98°C for 30 s, amplification cycles 98°C for 10 s, 68°C for 30 s, 72°C for 1 min 15 s (35 times), final extension 72°C for 5 min. Primers SR834 and SR835 were used to amplify a fragment of the *MARS1* coding sequence spanning from exon 16 to exon 28. Primers SR836 and SR837 were used to amplify the *GBLP* coding sequence, used as loading and positive control during the RT-PCR analysis. The sequence of the primers can be found in *Table 3*.

## Analysis of gene expression by quantitative polymerase chain reaction (qPCR)

For qPCR analyses, the cDNA was prepared as described above but the cDNA was diluted six to eight-fold in nuclease-free water prior to use. Primers to amplify the target transcripts are indicated in the *Table 3*. The qPCR reactions were carried out using the iQ Sybr green supermix as per manufacturer's instructions (Bio-Rad). To determine whether there was DNA contamination in the mix or whether there was primer dimer formation or misannealing, the same volume of the master mix, without cDNA, was added to a well in the 96-well plate. The raw Ct values were analyzed per the 'eleven golden rules', as previously described (*Udvardi et al., 2008*). *GBLP* was chosen as reference housekeeping transcript during normalization. Standard deviation was obtained for the 2–5 technical replicates and a minimum of 2 biological replicates were done per experiment, except for the experiment in *Figure 3—figure supplement 1B*, where only one biological replicate was analyzed.

The Phytozome v5.5 gene annotation for the target transcripts is as follows: *VIPP2* (Cre11.g468050.t1.2), *SULTR2* (Cre17.g723350.t1.2), *LHCBM9* (Cre06.g284200.t1.2), *HSP22F* (Cre14.g617400.t1.1), *SNOAL* (Cre11.g478100.t1.2), *LHCSR3.1* (Cre08.g367500.t1.1), *PSBS1* (Cre01.g016600.t1.2), *CPLD29* (Cre02.g088500.t1.1), *GBLP* (Cre06.g278222.t1.1).

## RNA-Seq: sample preparation and processing

Prior to RNA-seq analyses, the *mars1-1* strain was backcrossed to the wild-type CC-124 (Chlamydomonas Resource Center). A full tetrad was selected and analyzed as shown in *Figure 2E*. A wild-type and *MARS1* mutant progeny (CrPW8, indicated as E12, and CrPW9, indicated as F2 respectively, in *Figure 2E* and *Figure 2—figure supplement 2B*) were chosen for follow-up studies, based on the retention of the ClpP1 repressible system in their genetic background.

For each strain, two cultures (biological replicates) were inoculated in 30 ml liquid TAP medium using 50 ml Falcon tubes, starting from a fresh re-streak of cells propagated on TAP agar. These cultures were grown with mild agitation (150 rpm) on a shaker at around 22°C, under an illumination of 30–40 µmol photons $m^{-2} s^{-1}$ for about three-four days till they reach mid-late exponential phase (4–$7 \times 10^6$ cells $ml^{-1}$). Next, cells were diluted to about $1 \times 10^6$ cells $ml^{-1}$ in 30 ml of liquid TAP medium using 50 ml Falcon tubes and they were subjected to three different treatments: a) low light, i.e. they were kept at the same light intensity used during conditioning; b) HL, i.e. they were shifted to very high light (1200 µmol photons $m^{-2} s^{-1}$) for 40 min or 70 min; and c) ClpP1 repression, i.e. they were diluted in liquid TAP medium containing 100 µM thiamine and 40 ng $ml^{-1}$ vitamin $B_{12}$ and incubated for 68 hr at the same light intensity used during conditioning. Cells were then shifted to ice and quickly harvested by centrifugation at 3000 x g for 5 min at 4°C. Cell pellets were snap-frozen in liquid nitrogen and saved at −80°C till use. RNA extraction was carried as described in the previous section. The following extra measures were taken to ensure a complete removal of DNA contaminants: An additional round of in-solution Dnase I treatment was performed using 1 unit of Rnase-free Dnase I (Roche)/1 µg of total RNA at RT for 20 min in presence of 1 unit of recombinant ribonuclease inhibitors (RNaseOUT, Thermo Fisher Scientific). Next, the RNA was re-purified using the same extraction protocol described above.

Each total RNA preparation was ran on an Agilent Bioanalyzer RNA 6000 Nano chip for quantification and quality control. PolyA mRNAs were purified and RNAseq libraries were prepared using the Kapa mRNA HyperPrep kit (Roche, KK8540) following manufacturer's protocols. Libraries were pooled based on fragment analyzer concentrations. Sequencing was performed on Nextseq high-output flowcell, $1 \times 75$ bp run (Illumina).

## RNA-Seq: data analysis

We used a combination of publicly available tools and custom scripts for the processing of the raw demultiplexed Illumina sequencing data. Illumina adapter sequences were first trimmed off with TrimGalore! (version_0.4.1) (www.bioinformatics.babraham.ac.uk/projects/trim_galore) and contaminating ribosomal reads were removed by mapping against the Silva rRNA database using bbduk v37.32 (part of the BBTools suite, https://jgi.doe.gov/data-and-tools/bbtools/bb-tools-user-guide/). Quality control of raw and processed fastq files was performed using FastQC version 0.11.3 (https://www.bioinformatics.babraham.ac.uk/projects/fastqc/). The remaining reads were aligned to the unmasked *C. reinhardtii* genome (Phytozome, v5.5) using STAR v.2.5.3a (*Dobin et al., 2013*) and bam files were sorted with samtools 1.1 (*Li et al., 2009*). Count generation and downstream analysis were done in R (R project v3.4.0, www.R-project.org) using a combination of the packages Rsubreads (*Liao et al., 2013*) EdgeR (*Robinson et al., 2010*) plyr, ggplot2, gplots, and heatmap2. For differential expression analysis, genes with less than 0.5 counts per million reads in less than two samples were discarded, the data were fit to a negative binomial generalized linear model, and differential expression was determined using the quasi likelihood F-test with Benjamini-Hochberg correction of multiple testing in EdgeR. To subdivide genes into groups of *MARS1*-dependent and -independent genes, weakly-expressed genes (average RPKM <2.5 in at least two conditions) were discarded. Stress-responsive genes (defined as $log_2$-fold change >= |2| at FDR 0.001 upon treatment - high light, ClpP repression, or both as noted – in WT background) were considered *MARS1*-dependent when treatment in the *MARS1* mutant did not lead to a greater than 2-fold change in expression ($log_2$-fold change < |1|). *MARS1*-independent genes were defined as genes with an at least 4-

fold (log$_2$-fold change >= |2|) change upon treatment in the *mars1* background, in the same direction (up or down) as the response in WT cells. For analysis of functional categories in MapMan 10.0 (*Thimm et al., 2004*, *Usadel et al., 2005*) the *C. reinhardtii* v5.5 proteome (downloaded from Phytozome, v5.5, protein sequences from primary transcript only) was binned using Mercator4 (www.pla-bipd.de/portal/web/guest/mercator4) and *MARS1*-dependent and *MARS1*-independent gene subsets were mapped onto plant pathways in MapMan.

## Acknowledgements

We thank Jean David Rochaix for his generous donation of key reagents and critical feedback on this work, Olivier Vallon, Michael Schroda and Arthur Grossman for providing the ClpP1, Hsp22E/F and Sultr2 antibodies, respectively. We thank Fran Sanchez for preparing growth media for maintenance of our strains, Luke Mackinder, Nina Ivanova, Julia Wei for help with the genetic screen, and Lorenzo Costantino, Susan Dutcher, Stephane Gabilly, Ursula Goodenough, Robert Jinkerson, Yuval Kaye, Elif Karagoz, Xiaobo Li, Daniel Neusius, Jörg Nickelsen, Masayuki Onishi, Leif Palessen, Weronika Patena, Jirka Peschek, Shai Saroussi, Setsuko Wakao, Lan Wang and Tyler Wittkopp for invaluable advice in various aspects of this project. We thank Ryan Leib and Kratika Singhal who curated Mass-Spec analysis for this work and all Walter and Jonikas lab members for insightful discussions on this project.

## Additional information

### Funding

| Funder | Grant reference number | Author |
|---|---|---|
| European Molecular Biology Organization | LFT563-2013 | Silvia Ramundo |
| Schweizerischer Nationalfonds zur Förderung der Wissenschaftlichen Forschung | P300PA_161002 | Silvia Ramundo |
| Schweizerischer Nationalfonds zur Förderung der Wissenschaftlichen Forschung | P2GEP3_148531 | Silvia Ramundo |
| National Institutes of Health | DP2-GM-119137 | Martin C Jonikas |
| National Institutes of Health | R01GM032384 | Peter Walter |
| National Science Foundation | 2016218040 | Hannah Toutkoushian |
| National Science Foundation | 2012135643 | Mable Lam |
| National Science Foundation | MCB-1146621 | Martin C Jonikas |
| Howard Hughes Medical Institute | 55108535 | Martin C Jonikas |
| Howard Hughes Medical Institute | 826735-0012 | Peter Walter |

The funders had no role in study design, data collection and interpretation, or the decision to submit the work for publication.

### Author contributions

Karina Perlaza, Conceptualization, Investigation, Methodology, Validation, Formal analysis, Data Curation, Visualization, Writing – original draft preparation, Writing – review and editing; Hannah Toutkoushian, Investigation, Methodology, Validation, Data Curation, Visualization; Morgane Boone, Mable Lam, Masakazu Iwai, Software, Validation, Formal analysis, Resources, Data Curation, Visualization; Martin C Jonikas, Methodology, Resources, Funding acquisition; Peter Walter, Conceptualization, Resources, Writing – original draft preparation, Writing – review and editing, Supervision, Project administration, Funding acquisition; Silvia Ramundo, Conceptualization, Investigation,

Methodology, Validation, Resources, Data Curation, Writing – original draft preparation, Writing – review and editing, Visualization, Supervision, Project administration

**Author ORCIDs**
Karina Perlaza (iD) https://orcid.org/0000-0003-3297-884X
Hannah Toutkoushian (iD) https://orcid.org/0000-0002-7461-2005
Morgane Boone (iD) https://orcid.org/0000-0002-7807-5542
Masakazu Iwai (iD) https://orcid.org/0000-0002-0986-9015
Martin C Jonikas (iD) https://orcid.org/0000-0002-9519-6055
Peter Walter (iD) https://orcid.org/0000-0002-6849-708X
Silvia Ramundo (iD) https://orcid.org/0000-0002-2703-1398

**Decision letter and Author response**
Decision letter https://doi.org/10.7554/eLife.49577.sa1
Author response https://doi.org/10.7554/eLife.49577.sa2

## Additional files

### Supplementary files

• Source code 1. ImageJ macroscripts used to process and quantify YFP fluorescence and growth area of candidate mutant colonies on 384-well arrays.

• Supplementary file 1. Detailed information about peptide spectra sequences and post-translation modifications detected in Mars1 protein upon metronidazole treatment.

• Supplementary file 2. Average RPKM values for all Chlamydomonas transcripts in WT and *mars1-1* mutant upon high light exposure (for 40 min or 70 min) or ClpP1 down-regulation for 68 hr.

• Transparent reporting form

### Data availability

Unprocessed fastq files for whole genome and transcriptome sequencing samples were deposited in the Sequence Read Archive under the accession PRJNA529458 and PRJNA488111, respectively. Computer scripts used to analyze the transcriptome sequencing data and lists of *MARS1*-dependent and *MARS1*-independent genes are available through Figshare (https://figshare.com/s/992706a610ce6b71f03c and https://figshare.com/s/66417c2b28f3110b8077). All other data is available in the manuscript or the supplementary materials.

The following datasets were generated:

| Author(s) | Year | Dataset title | Dataset URL | Database and Identifier |
|---|---|---|---|---|
| Perlaza K, Toutkoushian H, Boone M, Lam M, Iwai M, Jonikas MC, Walter P, Ramundo S | 2019 | Whole genome sequencing of pooled progeny derived upon crossing *mars1-2* to WT | https://www.ncbi.nlm.nih.gov/bioproject/PRJNA529458 | NCBI Bioproject, PRJNA529458 |
| Perlaza K, Toutkoushian H, Boone M, Lam M, Iwai M, Jonikas MC, Walter P, Ramundo S | 2018 | Transcriptional stress responses in WT and *mars1-1* | https://www.ncbi.nlm.nih.gov/bioproject/PRJNA488111 | NCBI Bioproject, PRJNA488111 |

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
