## [Decision Letter]

Thank you for submitting your article "The Mars1 kinase confers photoprotection through signaling in the chloroplast unfolded protein response" for consideration by *eLife*. Your article has been reviewed by three peer reviewers, and the evaluation has been overseen by a Reviewing Editor and Christian Hardtke as the Senior Editor.

The reviewers have discussed the reviews with one another and the Reviewing Editor has drafted this decision to help you prepare a revised submission.

The reviewers were largely positive about your work and appreciated your elegant forward genetic approach in order to identify novel molecular players involved in cpUPR. However, they also identified some shortcomings that need to be addressed.

Essential revisions:

1) The title of your manuscript claims that MARS confers photoprotection, however, photosynthetic parameters were not presented. The reviewers ask you to either balance your discussion or to substantiate your claims by including photosynthetic parameters of wt and mars1 mutants under stress.

2) The reviewers also pinpointed the paper lack of experimental evidences for kinase activity and signalling transduction function of MARS1. Please either provide additional evidences or adjust the manuscript.

3) Your data suggests that MARS1 is involved in cpUPR, but it would be nice to discuss whether it could have additional, cpUPR-independent roles.

Please see below the additional comments of the reviewers, which may guide you to further strengthen your manuscript.

*Reviewer #1:*

This manuscript, to my knowledge, identifies and describes the first known component of the "chloroplast unfolded protein response" (cpUPR), a chloroplast to nucleus retrograde signaling pathway in the alga *Chlamydomonas reinhardtii*. Proteins often enter organellar compartments (i.e. ER, mitochondria and chloroplasts) in an unfolded state and must then be folded very quickly to avoid protein aggregation and cellular damage as a consequence. To prevent this type of damage from happening in chloroplasts cpUPR can be elicited. Stress such as excessive light can trigger cpUPR that will upregulate expression of nuclear-encoded chaperones, proteases and other factors aiding protein homeostasis. This manuscript describes an elegant high-throughput genetic screen (based on a riboswitch to turn off the chloroplast ClpP1 protease and activate cpUPR) in the *Chlamydomonas reinhardtii* algal model system to identify components of the signaling pathway that relays proteotoxic signals emanating from the chloroplast to the nucleus. The screen identifies the cytoplasmic kinase MARS1 (mutant affected in chloroplast-to-nucleus retrograde signaling). In addition, a dominant active allele of MARS1 is shown to constitutively induce cpUPR and alleviate photooxidative stress. The authors suggest that the pathway could be engineered in photosynthetic organisms to increase tolerance to chloroplast stress.

This is a very original manuscript that would cater well to a wide interest audience of *eLife*. The claims of the paper are well supported by a wealth of data often obtained using state of the art techniques. Furthermore, the paper is expertly written and the figures are of excellent quality.

Regarding MARS1 cellular localization: from the Results and Discussion section, Figure 3 and Figure 3—figure supplement 2A it was difficult to figure out the nature of MARS1-A (as well as -B, -C, -D). Could this be improved? Is it really necessary to mention the existence of a potential N-terminal chloroplast targeting signal if ATg(ii) would be the start codon? This was also somewhat unclear to me initially.

Stress in chloroplasts typically affects photosynthesis and this appears to be the case here as mars1 cells lacked photoprotection and exhibited accelerated photobleaching under highlight. It would therefore be really interesting to know whether and how photosynthetic parameters are affected by cpUPR and in the mars1 mutant. I'd consider such data standard in this type of study.

The discovery of MARS1 is exciting. It would be very nice to hear more about the authors' hypotheses on the mechanisms of MARS1-dependent signal transduction.

*Reviewer #2:*

This manuscript follows up on previous work performed by the corresponding author Ramundo that investigated how unfolded proteins in the chloroplast can be a retrograde stress signal in the green algae *Chlamydomonas*. Here, the authors attempt to identify signaling components of such a pathway by performing a clever and novel genetic screen in the same algae. They have constructed a strain with vitamin-inducible unfolded protein stress in the chloroplast and added a reporter construct with the YFP gene driven by the promoter of *VIPP2*, which is strongly induced by this particular retrograde signal. With this strain, they screened for mutants that failed to express YFP during the chloroplast unfolded protein response. Such mutants may be defective in their ability to send retrograde signals from stressed chloroplasts. Two true mutants were retrieved by the screen and both were alleles of *MARS1*, which encodes an uncharacterized cytoplasmic-localized kinase . The authors show that its kinase activity is required for the signals, and whole transcriptome analyses of *Chlamydomonas* show that this kinase is necessary to regulate about half of the genes controlled by the unfolded chloroplast protein response. Importantly, the authors show that the *MARS1* mutant cells are also sensitive to excess light and chloroplast hydrogen peroxide, demonstrating that the Mars1 protein is likely part of a physiologically relevant stress signal.

Altogether, this is very nicely performed work that should be of general interest to those in the field of abiotic stress signaling and organelles biology. I do however, think the manuscript feels a bit too compressed in parts and there are sections that could benefit from additional information, analysis, and detail (please see below).

Questions for the authors:

1) Can more information be provided about the Mars1 protein? What kind of domains does it have? Is it conserved among algae or plants? Are there any paralogues? etc. Are there any available bioinformatics data about expression at the transcript or protein level? Can any of this information provide clues to its function, localization, or evolutionary history?

2) For the complementation analysis, was flag-tagged Mars1 protein used? If so, this was not at all clear in the text. Again, more detail in the text will help to guide the reader through the manuscript.

3) The explanation of the localization studies were too brief. Please take the time to explain the experiment and how the conclusions were drawn. Also, was localization also tested under stress conditions? Perhaps MARS1 is mobile or uses alternative start sites.

*Reviewer #3:*

This manuscript provides novel information on a gene related to a signal transduction pathway for a so-called chloroplast unfolded protein response (cpUPR). The authors constructed an elegant reporter system for screening mutants that had defects in inducing *VIPP2* promoter activity upon suppression of the ClpP1 protease in the chloroplast of *Chlamydomonas*. Two mutants thus obtained were characterized in detail. In the case of the first mutant (*mars1-1*), a previously uncharacterized Ser/Thr kinase (Cre16.g692228) that they named Mars1 in this study was disrupted. Another mutant (mars1-2) also had a deletion encompassing *MARS1*. All mutants including two additional mutants from the CLiP library showed a similar phenotype. Moreover, they confirmed the lack of Mars1 transcript expression in the mutants and the phenotypic rescue of the mutants by the cDNA. The *mars1* phenotype was sensitive to a mutation in a possible Ser/Thr kinase domain. The *mars1* mutant did not express Vipp2 and Hsp22E/F and bleached under HL or under normal light in the presence of metronidazole. The mutants affected the expression of many high light-inducible genes, which seem to belong to a distinct cluster from that previously reported as NPQ related functions. Based on these results, the authors propose that a retrograde signal from the chloroplast activates the cytoplasmic kinase Mars1 upon chloroplast proteotoxic stress, which in turn orchestrates the cpUPR transcriptional program.

The experiments here were carefully performed, the presentations are mostly clear, and the methods are meticulously described. This is a solid study of a new possibly interesting kinase. On the whole, however, I got the impression that the research is just premature. I have a suggestion if this manuscript is eventually published in *eLife*.

The nature of proteotoxic stress studies here should be more clarified.

As shown by the result, this cpUPR event led by Mars1 was not affected by tunicamycin, indicating it is independent of the stress due to the accumulation of premature proteins (UPR). The authors found it was promoted by either suppression of ClpP1 or HL stress in the chloroplast, suggesting cpUPR is rather dependent on the stress due to the accumulation of degraded proteins in the chloroplast. The two types of stress seem to be fundamentally distinct. In fact, if the current result is showing a mechanical relationship between UPR and cpUPR, that could be of broader interest. If the Mars1-dependent genes were involved in cpUPR, they would be induced under the conditions when ClpP1 was OFF and HL was illuminated. However, there were 43 Mars1-dependent genes among HL inducible genes including *SIR*, *CHLM*, *DSS1*, etc. that were not expressed under ClpP1 OFF conditions (Figure 5C). How to explain this?

The authors are encouraged to fill the fatal link between the proteotoxic stress and Mars1-dependent phosphorylation, preferably by determining the retrograde signal and/or the partner of the Mar1 kinase.

---

## [Author Response]

Essential revisions:1) The title of your manuscript claims that Mars1 confers photoprotection, however, photosynthetic parameters were not presented. The reviewers ask you to either balance your discussion or to substantiate your claims by including photosynthetic parameters of wt and mars1 mutants under stress.

We have edited our manuscript to balance our discussion on photoprotection (subsection “Conclusions”). In brief, we provide clear evidence that Mars1 mitigates photooxidative stress and delays photobleaching. Moreover, loss of Mars1 does not impair expression of genes involved in non-photochemical quenching (Figure 5C), suggesting that photoprotection is conferred by a different mechanism. However, given the large number of photoprotective strategies that plants employ to cope with such stresses, determining the molecular mechanism(s) by which this occurs is not a straightforward undertaking that is beyond the scope of this work.

2) The reviewers also pinpointed the paper lack of experimental evidences for kinase activity and signalling transduction function of Mars1. Please, either provide additional evidences or adjust the manuscript.

We have obtained mass spectrometry data, which show that, as expected, Mars1’s activation loop is phosphorylated in stress (at S1888). This phosphorylation event is dependent on Mars1’s kinase activity and absent in catalytically-dead Mars1. We have now included these findings in Supplementary file 1 and Author response image 1.

**Author response image 1. respfig1:** Multiple sequence alignment of Mars1 kinase domain (1638-2019 aa) and the five proteins selected by RaptorX as PDB templates to predict Mars1 tertiary structure. The canonical ATP-binding loop, catalytic triad and EF-helix regions are highlighted by green rectangular frames. The red asterisk indicated the phosphoserine (pS) residue identified in Mars1 activation loop by Mass-Spec analysis. This multiple sequence alignment was generated by ClustalW (Thompson et al., 1994) and its results were visualized using MView (Brown et al., 1998).

Moreover, all important structural motifs found in canonical kinases, including the catalytic triad, the ATP binding loop, the activation loop, as well as all canonical secondary structure elements are conserved in the Mars1 kinase domain (for details, see Author response image 1 and Author response image 2). While we agree that an in vitro biochemical assay would be the most direct way to show activity, purifying Mars1 has proved challenging due to its large size. Despite considerable efforts, we are unable to provide such results in the short-term.

**Author response image 2. respfig2:** Mars1 tertiary structure prediction. (A) Tertiary structural alignment of Mars1 (light blue) and human VEGFR2 kinase (magenta) in complex with a small molecule inhibitor, Axitinib (gold) (PDB ID: 4agc). 4agc was selected by the RaptorX web server (Kallberg et al., 2012) as best PDB template (p-value 5.45e-10) to model Mars1 tertiary structure. (B) Side-by-side comparison of ATP-binding loop, catalytic triad and EF-helix regions in Mars1 (light blue) and human VEGFR2 (magenta). The different conformation of the phenylalanine in the DFG region of VEGFR2 is due to the presence of a type I kinase inhibitor (Axitinib) (gold), which occupies the front pocket region, the adenine-binding area and the DFG-motif of this enzyme.

3) Your data suggests that Mars1 is involved in cpUPR, but it would be nice to discuss whether it could have additional, cpUPR-independent roles.

We have no indication that Mars1 is involved in any other pathway in addition to the cpUPR.

Please, see below the additional comments of the reviewers, which may guide you to further strengthen your manuscript.Reviewer #1:[…] Regarding Mars1 cellular localization: from the Results and Discussion section, Figure 3 and Figure 3—figure supplement 2A it was difficult to figure out the nature of *MARS1-A*. Could this be improved? Is it really necessary to mention the existence of a potential N-terminal chloroplast targeting signal if ATg(ii) would be the start codon? This was also somewhat unclear to me initially.

We now include mass spectrometry data, which strongly fits with the hypothesis that the translation start of *MARS1* transcript is ATG*(i)* and not the Phytozome-annotated ATG*(ii)*. These data show that multiple peptide spectra (including one with a phosphorylated Serine, S69) can be detected from the extended N-terminus region translated when ATG*(i)* is used as the start codon (Supplementary file 1). Accordingly, we have updated the manuscript (Results and Discussion, seventh paragraph) and significantly simplified Figure 3A and its respective legend.

Stress in chloroplasts typically affects photosynthesis and this appears to be the case here as mars1 cells lacked photoprotection and exhibited accelerated photobleaching under highlight. It would therefore be really interesting to know whether and how photosynthetic parameters are affected by cpUPR and in the mars1 mutant. I'd consider such data standard in this type of study.

Please refer to our response to essential revision (1) above.

The discovery of Mars1 is exciting. It would be very nice to hear more about the authors' hypotheses on the mechanisms of Mars1-dependent signal transduction.

We entirely agree. We are very excited to elucidate, at the mechanistic level, how Mars1 fits into a cpUPR signaling network and are actively pursuing the isolation and characterization of additional players. Currently, we can envision many signaling mechanisms, but feel that such discussion would remain merely hypothetical. Ideally, we prefer to stay agnostic on this topic.

Reviewer #2:[…] Altogether, this is very nicely performed work that should be of general interest to those in the field of abiotic stress signaling and organelles biology. I do however, think the manuscript feels a bit too compressed in parts and there are sections that could benefit from additional information, analysis, and detail (please see below).Questions for the authors:1) Can more information be provided about the Mars1 protein? What kind of domains does it have? Is it conserved among algae or plants? Are there any paralogues? etc. Are there any available bioinformatics data about expression at the transcript or protein level? Can any of this information provide clues to its function, localization, or evolutionary history?

We have now included circadian gene expression data for Mars1 in the Discussion. These data align with our proposal of its photoprotective function (Results and Discussion, thirteenth paragraph; Figure 5—figure supplement 5A-B). With the exception of the kinase domain at the C-terminus, no other known domain can be recognized in Mars1. Sequence alignments fail to identify Mars1 in higher plants and the evolutionary lineage of this protein remains to be deciphered.

2) For the complementation analysis, was flag-tagged Mars1protein used? If so, this was not at all clear in the text. Again, more detail in the text will help to guide the reader through the manuscript.

We have clarified this point. All the rescue experiments were performed using one of our Flag-tagged *MARS1* transgenes (namely, *MARS1-A* or *D*, each of them containing the triple Flag epitope at a different location). This information is provided in the text (Results and Discussion, seventh paragraph) and now clearly highlighted in Figure 3A.

3) The explanation of the localization studies were too brief. Please take the time to explain the experiment and how the conclusions were drawn. Also, was localization also tested under stress conditions? Perhaps Mars1 is mobile or uses alternative start sites.

See our first response to reviewer #1. We have not tested Mars1 localization upon stress but mass spectrometry data suggest that the same start site, ATG*(i)*, is used in presence of stress. We have added this information to the text (Results and Discussion, eleventh paragraph) and we now provide the sequences of the Mars1 spectra peptides in Supplementary file 1.

Reviewer #3:[…] This is a solid study of a new possibly interesting kinase. On the whole, however, I got the impression that the research is just premature. I have a suggestion if this manuscript is eventually published in eLife.The nature of proteotoxic stress studies here should be more clarified.As shown by the result, this cpUPR event led by Mars1 was not affected by tunicamycin, indicating it is independent of the stress due to the accumulation of premature proteins (UPR). The authors found it was promoted by either suppression of ClpP1 or HL stress in the chloroplast, suggesting cpUPR is rather dependent on the stress due to the accumulation of degraded proteins in the chloroplast. The two types of stress seem to be fundamentally distinct. In fact, if the current result is showing a mechanical relationship between UPR and cpUPR, that could be of broader interest. If the Mars1-dependent genes were involved in cpUPR, they would be induced under the conditions when ClpP1 was OFF and HL was illuminated. However, there were 43 Mars1-dependent genes among HL inducible genes including *SIR*, *CHLM*, *DSS1*, etc. that were not expressed under ClpP1 OFF conditions (Figure 5C). How to explain this?

To clarify: the cpUPR is a fundamentally different pathway from the erUPR (just as mtUPR and erUPR are distinct), and our data do not indicate any relationship between these two pathways.

As for the RNAseq analysis, the goal of this experiment was to identify a core set of cpUPR target genes that are consistently differentially expressed in all tested cpUPR inducing conditions, namely ClpP1 knock-down and high light. To this aim, we applied a rather stringent but arbitrary cut-off (4-fold change, p<0.001), when deciding which genes are differentially expressed in each or both conditions.

The 43 genes annotated as *MARS1*-dependent only upon high light are also induced upon ClpP1 knockdown (see example in Author response table 1). However, they did not meet the cut-off criteria for being considered differentially expressed in this condition.

We realize that this description of the gene expression data (Figure 5C) may have led to some confusion and have adjusted it accordingly. Furthermore, we now include an Excel file (Supplementary file 2) containing all the average RPKM values of all genes in all conditions, so that these data can be readily accessed by any reader.

WT (average RPKM)*mars1* (average RPKM)Transcript IDSymbolunstressedHL 40”HL 70”ClpP1 OFFunstressedHL 40”HL 70”ClpP1 OFFCre12.g498550.t1.1*CHLM*65396333215789917553Cre02.g095093.t1.1*DSS1*3619993898551049758Cre16.g693202.t1.2*SIR*361029961994330218

The authors are encouraged to fill the fatal link between the proteotoxic stress and Mars1-dependent phosphorylation, preferably by determining the retrograde signal and/or the partner of the Mar1 kinase.

We are eager to fill the gap between Mars1 activity and proteotoxic stress. However, we think that the identification of Mars1 as a pivotal member of a new communication route is already a crucial breakthrough in the field.